**Analysis**

# Influence of different sample preparation approaches on proteoform identification by top-down proteomics

Philipp T. Kaulich [1], Kyowon Jeong [2,3], Oliver Kohlbacher [2,3,4] & Andreas Tholey [1] ✉

Top-down proteomics using mass spectrometry facilitates the identification of intact proteoforms, that is, all molecular forms of proteins. Multiple past advances have lead to the development of numerous sample preparation workflows. Here we systematically investigated the influence of different sample preparation steps on proteoform and protein identifications, including cell lysis, reduction and alkylation, proteoform enrichment, purification and fractionation. We found that all steps in sample preparation influence the subset of proteoforms identified (for example, their number, confidence, physicochemical properties and artificially generated modifications). The various sample preparation strategies resulted in complementary identifications, substantially increasing the proteome coverage. Overall, we identified 13,975 proteoforms from 2,720 proteins of human Caco-2 cells. The results presented can serve as suggestions for designing and adapting top-down proteomics sample preparation strategies to particular research questions. Moreover, we expect that the sampling bias and modifications identified at the intact protein level will also be useful in improving bottom-up proteomics approaches.

Proteoforms are different molecular forms of a given protein, including genetic variants, splice variants and co- and posttranslational modifications (PTMs)[1]. Since different proteoforms can have considerably different biological functions, their identification, characterization and quantification are crucial to understanding molecular mechanisms[2]. In contrast to bottom-up proteomics (BUP), where the proteins are digested into peptides and proteins are inferred, the identification of intact proteoforms by top-down proteomics (TDP) can theoretically reveal the entire complexity of the proteome[3].

Despite major advances made in the recent past[4–10], the proteome-wide identification of proteoforms by TDP is still hampered by several challenges at all stages of the analysis, that is, sample preparation, proteoform separation, mass spectrometric analysis and database search[11–13]. These challenges become more relevant with the increasing size of proteoforms. For example, the sensitivity in electrospray-mass spectrometry inherently decreases with increasing proteoform size since the intensity is split into multiple charge states and isotopologues[14]. Additional challenges in TDP are broad chromatographic peak widths, overlapping signals and complex tandem mass spectrometry (MS/MS) spectra. TDP currently also faces an upper mass limit of approximately 30 kDa (refs. 15,16). Consequently, the enrichment of smaller proteoforms or the depletion of larger proteoforms is a common step in TDP sample preparation before liquid chromatography with MS/MS (LC–MS/MS) analysis.

A typical TDP sample preparation workflow for analyzing cellular proteoforms includes cell lysis, sample cleanup, isolation of

[1]Systematic Proteome Research and Bioanalytics, Institute for Experimental Medicine, Christian-Albrechts-Universität zu Kiel, Kiel, Germany. [2]Applied Bioinformatics, Computer Science Department, University of Tübingen, Tübingen, Germany. [3]Institute for Bioinformatics and Medical Informatics, University of Tübingen, Tübingen, Germany. [4]Translational Bioinformatics, University Hospital Tübingen, Tübingen, Germany. ✉e-mail: a.tholey@iem.uni-kiel.de

proteoforms within a suitable mass range and proteoform purification. Moreover, numerous TDP protocols that involve disulfide reduction and alkylation are available[4,5,8,17–19]. Cell lysis is performed to extract the proteoforms and needs to maintain their biological state (for example, by preventing enzymatic activity)[20]. For this, various lysis buffers have been reported in TDP studies, differing in pH, ionic strength, salts, protease inhibitors and detergents[21–23].

A plethora of methodologies have been presented for the isolation and purification of suitable proteoforms, such as gel-based approaches[19,24], solid-phase extraction (SPE)[4,25,26], depletion methods, molecular weight cutoff (MWCO) filters[17,18] and size-exclusion chromatography (SEC)[27,28].

In most TDP studies, proteoforms are separated by multidimensional fractionation schemes. For this purpose, LC-based approaches, such as reversed-phase chromatography (for example, high/low pH and low/low pH separation schemes)[29–31] or SEC[4,28] and gel-based fractionation approaches[19,24], are typically used. Enriching and fractionating proteoforms using gel-eluted liquid fraction entrapment electrophoresis (GELFrEE) have been state-of-the-art in the TDP community for many years[24]; a method that is gradually being replaced by polyacrylamide-gel-based prefractionation for analysis of intact proteoforms and protein complexes by mass spectrometry (PEPPI–MS)[19], in which the proteins are separated via SDS–PAGE and eluted from the gel by passive elution. Furthermore, gas-phase fractionation strategies using high-field asymmetric waveform ion mobility spectrometry (FAIMS) have been recently added as an additional separation technique for TDP[17,22,32–35].

Numerous different TDP sample preparation protocols combining the above-mentioned approaches have been described. Here we systematically examined the influence of various TDP sample preparation steps on the identifications of proteoforms from human Caco-2 cells (Fig. 1). Specifically, the influence of different lysis conditions, proteoform reduction and alkylation, various proteoform enrichment approaches and multidimensional separation schemes on the number, confidence, artificially introduced modifications and physicochemical properties of identifications were investigated. The results can provide suggestions for tailoring TDP sample preparation strategies to address the needs of specific research questions.

## Results

### LC–MS/MS workflow

To allow a fair comparison of the influence of various TDP sample preparation steps on proteoform identifications, an appropriate LC–MS/MS workflow was established. Two LC–FAIMS–MS methods using internal compensation voltage stepping to target proteoforms below (low-molecular-weight (LMW) method) and above (high-molecular-weight (HMW) method) 15 kDa were established (Supplementary Table 1)[32]. ProSightPD was used for proteoform identification with strict filtering criteria, including a <1% FDR cutoff and exclusion of proteoforms with a C-score below 40, to minimize false-positive identifications[36].

A detailed description of the LC–FAIMS–MS/MS workflow optimization is given in the Supplementary Results and Supplementary Figs. 1–7. In brief, the LMW and HMW methods turned out to be complementary and together cover a wide proteoform mass range, enabling the identification of proteoforms up to approximately 45 kDa. Three replicate injections per sample maximized the number of identifications while being economical with measurement time. Notably, the replicates yield highly reproducible identifications regarding proteoform count, confidence and physicochemical properties (Supplementary Figs. 5 and 6). The influence of the injection amount (30–1,200 ng) on proteoform identifications was examined, showing that with increasing injection amounts, a higher number and confidence of proteoform identification was achieved (Supplementary Fig. 7). Thus, to ensure a fair comparison of different sample preparation strategies,

approximately the same protein amount, based on the total ion count, was injected.

### Cell lysis

To investigate the influence of various lysis solutions on proteoform and protein identifications by TDP, Caco-2 cells were lysed in (1) phosphate-buffered saline (PBS), (2) ammonium bicarbonate-buffered urea (Urea–ABC), (3) guanidinium hydrochloride (GndHCl), (4) Tris-buffered sodium dodecyl sulfate (SDS–Tris), (5) acidic acetonitrile-water solution containing sodium chloride (ACN–NaCl) and (6) triethylammonium bicarbonate-buffered ACN-water solution (ACN–TEAB).

The largest number of identified proteoforms could obtained after GndHCl and ACN–TEAB lysis (Fig. 2a,b). However, a detailed inspection of the proteoforms revealed that mainly truncated proteoforms were detected, and only a few full-length proteoforms were identified. The truncated proteoforms were matched to their full-length sequences deposited in the database, and the potential truncation sites were determined (Supplementary Fig. 8). After PBS, SDS–Tris, Urea–ABC and ACN–NaCl lysis, a diverse distribution of potential truncation sites was observed, with a slight preference for the aspartate–proline bond. However, the proteoforms identified after GndHCl lysis showed a clear bias toward hydrolysis of peptide bonds C terminal to aspartate residues, especially of the aspartate–proline bond. Notably, elevated temperatures and an acidic environment can facilitate the hydrolysis of peptide bonds C terminal to aspartate residues[37,38]. Thus, due to the acidity of the unbuffered GndHCl solution, we hypothesized that these truncated proteoforms are, in part, artificial chemical hydrolysis products[39]. However, in vivo cleavage of the aspartate–proline bond has also been reported[40].

Next, the physicochemical properties of the identified proteoforms were investigated. The largest median mass of the proteoforms was obtained after lysis with PBS (11.8 kDa) and SDS–Tris (10.3 kDa), whereas lysis with GndHCl (7.4 kDa) and urea (7.9 kDa) resulted in the identification of smaller proteoforms (Fig. 2c). This observation agrees with a previous study investigating different lysis conditions of bacteria, which also concluded that lysis with urea leads to the identification of mainly small proteoforms[21]. As expected, the smallest proteoforms were identified after lysis with ACN–NaCl (7.2 kDa) and ACN–TEAB (4.6 kDa), since the ACN lysis or depletion protocol has been initially developed for enriching small proteins[41]. Notably, the presence of small proteoforms typically leads to a high number of identifications due to their inherently higher sensitivity in MS detection compared to larger proteoforms[14].

The isoelectric point (pI) of the proteoforms showed a similar distribution after PBS, SDS–Tris and Urea–ABC lysis, with a bias toward more basic proteoforms (pI > 9) (Fig. 2d). The lysis with ACN–TEAB led to a bias toward acidic proteoforms, whereas the lysis with ACN–NaCl resulted in more basic proteoforms (for example, histones). The bias toward specific proteoform subgroups can possibly be explained by the lower solubility of proteoforms in solutions with a pH close to their pI. For example, in neutral to acidic lysis solutions, proteoforms with alkaline pI are tendentially higher charged, potentially resulting in an improved extraction efficiency compared to proteoforms with neutral or acidic pI. Note that due to the complexity of the physicochemical properties of the molecules forming the proteome, there are multiple, sometimes interdependent, factors influencing proteoform solubilities. Thus, factors such as ion strengths and polarity of the solvents and their influence on the structures or interactions can possibly explain the observed effects, too. The ACN lysis buffers extracted proteoforms within a wide range of GRAVY scores (Fig. 2e). The lysis with the chaotropic salts urea and GndHCl showed a bias toward more hydrophobic proteins compared to PBS and SDS–Tris lysis.

The most similar identifications regarding proteoform and protein overlap coefficient (on average, ~56 and 73%, respectively) were obtained between the Urea–ABC, SDS–Tris and PBS lysis buffers

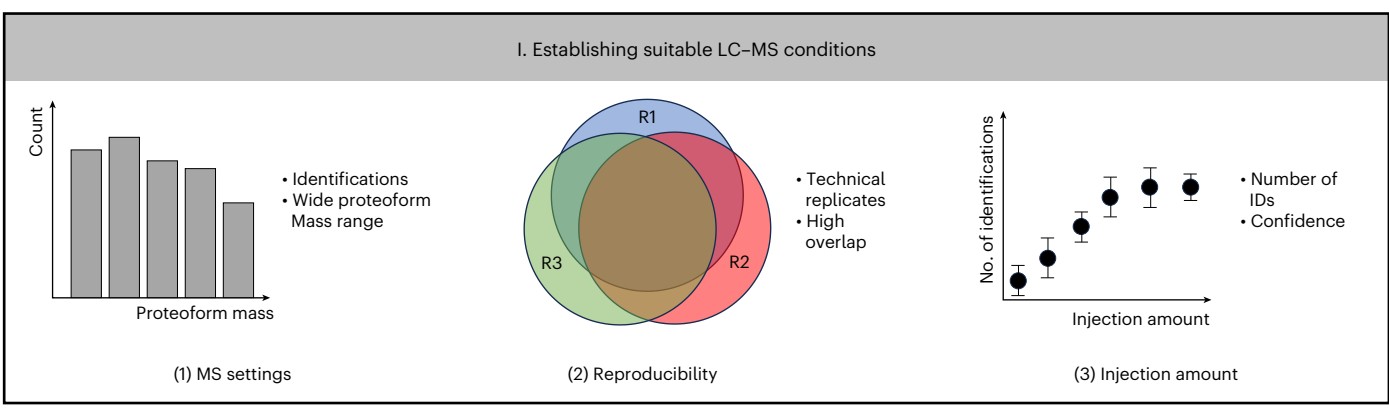

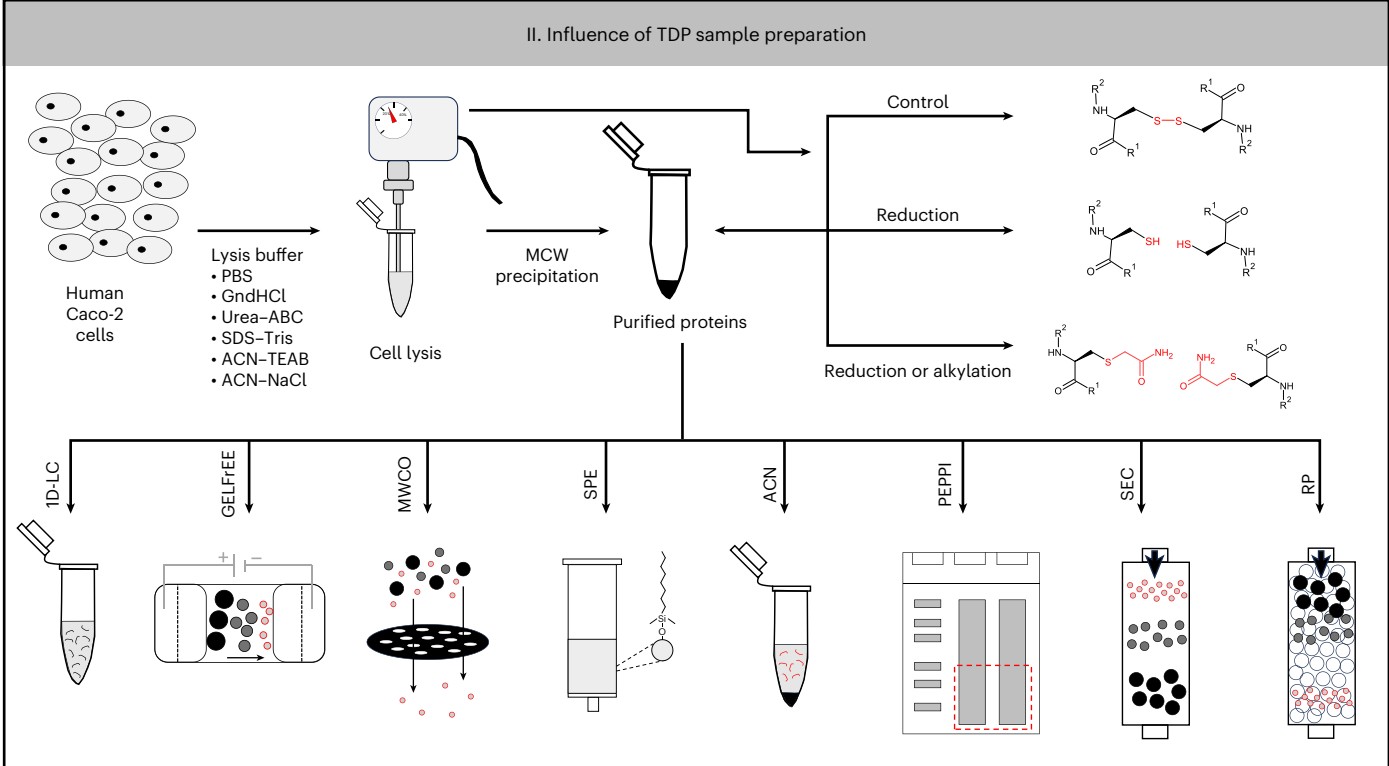

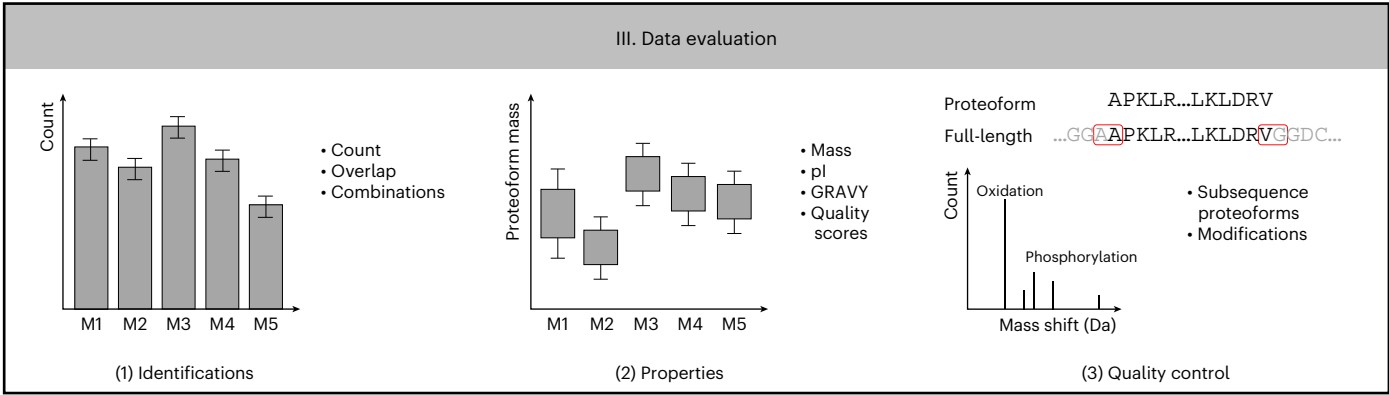

**Fig. 1 | Overview of the study design.** Upper panel: a suitable LC–MS/MS workflow was established, including optimizing the MS settings, number of replicate measurements, injection amount and reproducibility. Middle panel: the established LC–MS/MS workflow was used to investigate the influence of various sample preparation steps on proteoform identification, including different lysis conditions, proteoform reduction/alkylation, various prefractionation strategies to enrich proteoforms smaller than 30 kDa and multidimensional separation schemes. Lower panel: data analysis with ProSightPD regarding the proteoform count, reproducibility, complementarity, physicochemical properties and artificially introduced modifications. ACN, ACN depletion; RP, two-dimensional reversed-phase low/low pH proteoform fractionation approach.

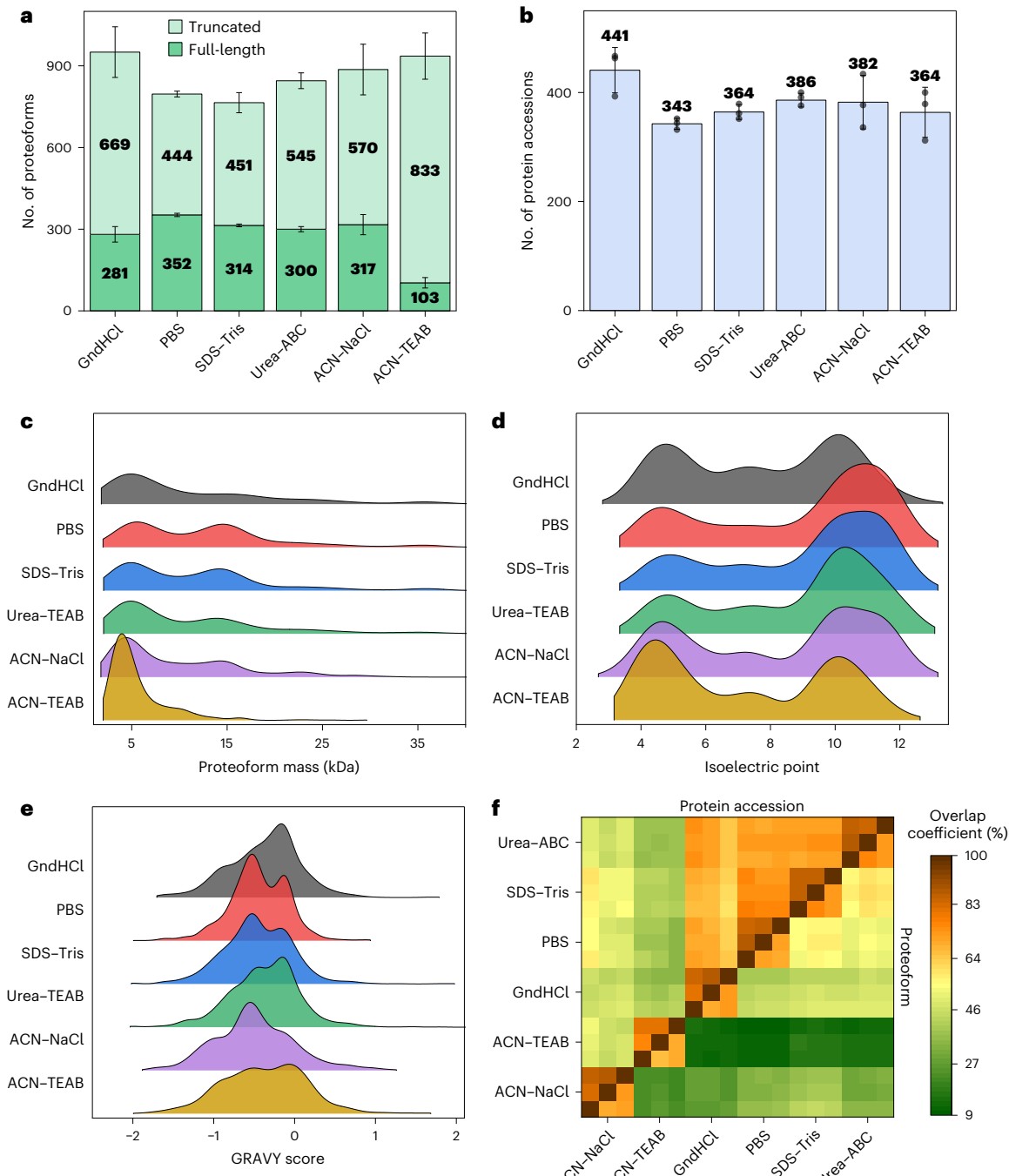

**Fig. 2 | Influence of the cell lysis conditions on proteoform identifications.** **a**,**b**, Number of identified proteoforms (**a**) and protein accessions (**b**) (*n* = 3 replicates of independently performed sample preparations, average ± standard derivation). Full-length proteoforms are those deposited in the proteoform database, including start methionine excision and previously described proteolytic-processed proteoforms, for example, with cleaved signal peptides. By contrast, truncated proteoforms refer to all subsequence proteoforms not defined in the database. **c**–**e**, Distribution of proteoform mass (**c**), isoelectric point (**d**) and GRAVY score (**e**). **f**, Overlap coefficients of the proteoforms and proteins identified in the replicates from the various cell lysis conditions.

(Fig. 2f), maintaining a neutral or alkalic pH during the cell lysis. In contrast, after lysis with GndHCl, the overlap coefficients with the neutral or alkalic lysis conditions were much lower (45% on proteoform and 67% on protein level). Furthermore, the lysis with ACN–TEAB and ACN–NaCl resulted in very low overlap coefficients compared to the other lysis conditions (23% on proteoform and 45% on protein level), that is, providing high complementarity.

For the detection of artificially introduced modifications, the raw data were deconvolved with FLASHDeconv[42] and analyzed by MSTopDiff[43] (Supplementary Fig. 9). Multiple oxidation events were observed in all samples, with the highest abundance after GndHCl lysis. After ACN–TEAB lysis, a mass shift that could be assigned to 4-(2-aminoethyl)benzenesulfonyl reaction products was observed, a serine protease inhibitor within the protease inhibitor mix and a common artifact in proteomics studies[44]. Moreover, the various lysis conditions influenced the number of identified posttranslationally modified proteoforms (Supplementary Table 2).

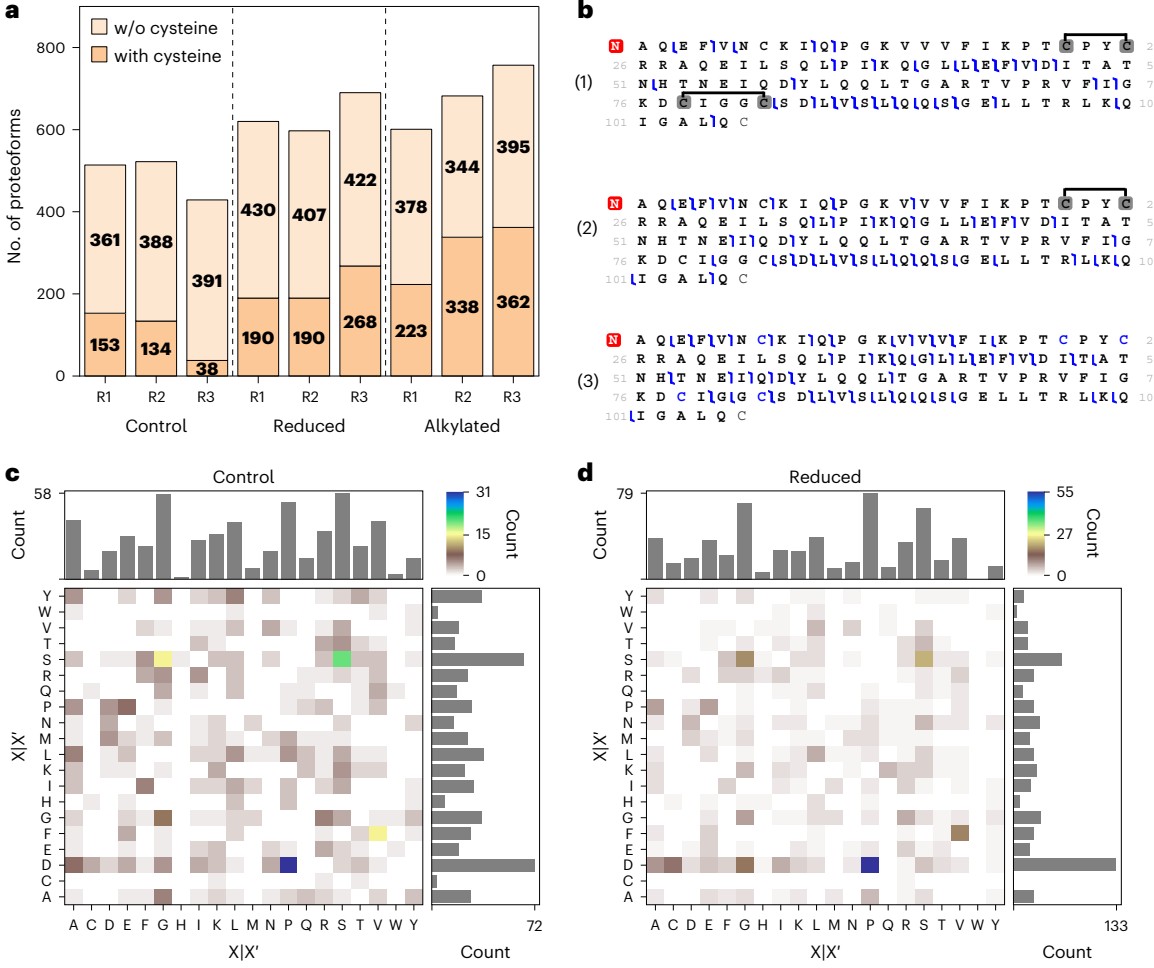

**Fig. 3 | Influence of proteoform reduction or reduction/alkylation on proteoform identification. a**, Number of identified proteoforms with or without cysteine residues. **b**, Fragment maps of selected proteoforms from Glutaredoxin-1 (P35754) identified in the (1) control, (2) reduced and (3) reduced/alkylated samples. The blue brackets represent b-/y-ions identified after CID fragmentation. The cysteine residues highlighted in gray indicate disulfide bridges and those in blue are carbamidomethylated cysteines. **c,d**, Analysis of truncated proteoforms in the control (**c**) and reduced sample (**d**). The two-dimensional histograms display the potential truncation sites of truncated proteoforms. The amino acids N and C terminal of the potential truncation sites are denoted as X and X', respectively.

## Proteoform reduction and alkylation

To investigate the influence of cysteine reduction and/or alkylation on proteoform identifications, Caco-2 cells were lysed and the proteoforms were subjected in triplicates to reduction with or without subsequent alkylation. An untreated sample was used as a control. Analysis of the raw files using MSTopDiff revealed no substantial occurrence of artificial mass shifts in the three samples, demonstrating that no overalkylation occurred (Supplementary Fig. 10).

After reduction or reduction and alkylation of the sample, a considerably elevated number of proteoforms was identified ($636 \pm 48$ and $680 \pm 78$, respectively, $n = 3$) compared to the control ($488 \pm 52$), with a major increase in the number of cysteine-containing proteoforms (Fig. 3a). In the control sample, several proteoforms with one or multiple disulfide bridges were identified, such as a proteoform from Glutaredoxin-1 containing five cysteine residues with two disulfide bridges (Fig. 3b). Furthermore, the linkage of the cysteines could be elucidated by the presence of fragment ions between unconnected cysteine residues and the absence of fragment ions between two connected cysteine residues. The same proteoform sequence was identified in the reduced or reduced/alkylated sample with carbamidomethylated cysteine residues, with a considerably higher residue cleavage (42%) compared to the untreated sample (29%). However,

the information about the disulfides was lost. When the sample was reduced without subsequent alkylation, the proteoform sequence was identified with two and one disulfide bridge, respectively, likely due to insufficient reduction or re-formation of the disulfide bridges during sample preparation.

The analysis of the potential truncation sites revealed that many more proteoforms originating from hydrolysis of peptide bonds C terminal to aspartate residues were identified after reduction or reduction/alkylation compared to the control (Fig. 3c,d and Supplementary Fig. 10). This observation might be attributed to the 1-h incubation at 50 °C since elevated temperatures can facilitate the hydrolysis of peptide bonds C terminal to aspartate residues[45]. Notably, although most TDP studies have performed reduction at elevated temperatures[8,19], this step can also be performed at room temperature[46], which potentially reduces the formation of artificially truncated proteoforms.

## Proteoform enrichment, depletion and purification

Due to the current upper mass limit in TDP, there is a common practice of isolating proteoforms smaller than approximately 30 kDa before LC−MS/MS analysis. Hence, we designed an experiment to investigate the influence of different proteoform prefractionation

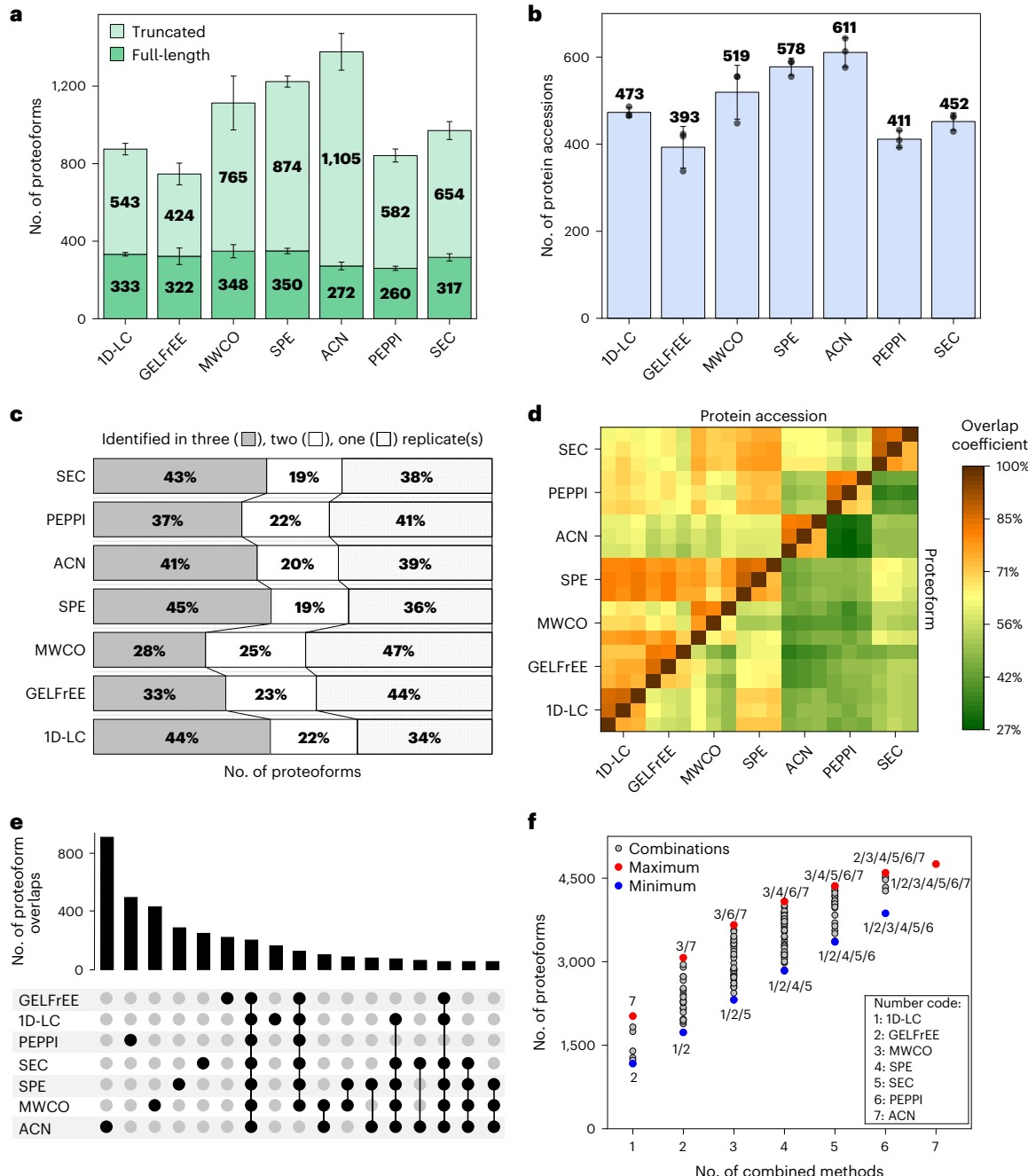

**Fig. 4 | Influence of various sample preparation strategies on the identification of proteoforms and proteins. a,b,** Number of identified full-length and truncated proteoforms (**a**) and protein accessions (*n* = 3 replicates of independently performed sample preparations, average ± standard derivation) (**b**). **c,** Reproducibility of the various approaches, showing the percentage of proteoforms identified in one, two or three replicates. **d,** Overlap coefficients between the replicates of all approaches regarding the identified proteins

and proteoforms. **e,** UpSet plot of proteoforms identified with the various approaches (minimum 50 proteoforms). **f,** Number of identified proteoforms when combining multiple database search results from the different approaches. All raw files were analyzed together in a multiconsensus analysis, and the number of proteoforms identified in a combination of various sample preparations was calculated. MWCO, 30-kDa filter; ACN, acidic (TFA/NaCl) ACN depletion; SPE, C18 material; PEPPI with subsequent MCW precipitation.

strategies, including MWCO filters (30 kDa), PEPPI (with subsequent proteoform purification by methanol-chloroform-water (MCW) precipitation), GELFrEE, SEC, SPE (C18 material) and acidic (trifluoroacetic acid (TFA)/NaCl) ACN depletion (Fig. 1), on TDP proteoform and protein identifications. Notably, many of these methods, such as SPE and MWCO, have also been used to desalt the sample rather than enrich small proteoforms[17,28,47]. An untreated sample was resuspended in LC–MS loading buffer as a control (denoted 1D-LC in the following). For quality control of the sample preparations and

the acquired data, see Supplementary Results and Supplementary Figs. 11–13.

**Identifications, reproducibility and complementarity**
Compared with the 1D-LC control, more proteoforms were identified after ACN depletion, SPE, MWCO filter and SEC (Fig. 4a). While the numbers of annotated proteoforms are all in the same range, the number of truncated proteoforms differed considerably between the various approaches. After the gel-based separations, many truncated

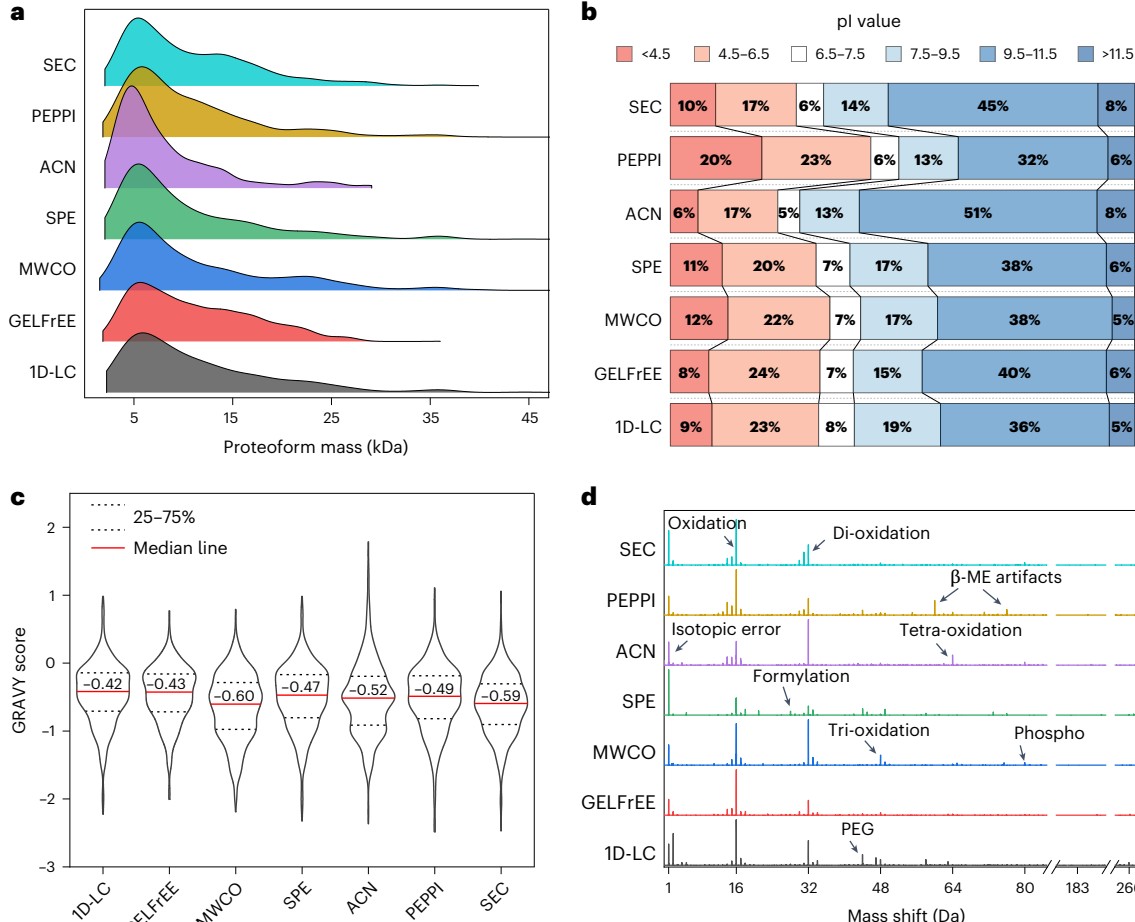

**Fig. 5 | Physicochemical properties of the identified proteoforms in dependence on the proteoform isolation strategy. a–c**, Distribution of the proteoform mass (**a**), isoelectric point (pI) (**b**) and GRAVY score (**c**). **d**, Detection of modifications using MSTopDiff after deconvolution of a randomly selected raw file from each approach with FLASHDeconv. Shown are the intensity × count histograms. β-ME, β-mercaptoethanol; PEG, polyethylene glycol contamination.

proteoforms derived from the hydrolysis of peptide bonds C terminal to aspartate residues (Supplementary Fig. 14), which may be explained by the heating step in the sample buffer before the separation. Besides that, the potential truncation sites showed no apparent bias for specific peptide bonds and differed only slightly between the various approaches. The number of identified proteins correlated with the total number of proteoforms (Fig. 4b).

The best reproducibility regarding proteoform identifications was achieved using SPE, whereas the lowest reproducibility was obtained with the MWCO filter and the GELFrEE approach (Fig. 4c). The 1D-LC, GELFrEE and SPE samples had the highest overlap coefficients between each other (~70% on proteoform level): that is, the lowest complementarity (Fig. 4d). In contrast, the highest complementarity to all other approaches was observed after ACN depletion (30–50%). Many proteoforms (56%, $n$ = 2,703) and proteins (38%, $n$ = 536) were uniquely identified in individual approaches (Fig. 4e and Supplementary Fig. 15). Compared to that, only 198 proteoforms (4%) and 163 proteins (12%) were identified in all of the seven investigated approaches, demonstrating high complementarity.

We next investigated the number of identified proteoforms when the search results of multiple sample preparation approaches were combined (Fig. 4f). The more methods were combined, the higher the number of identifications; however, the gain in identifications decreased as the number of combined approaches increased. Notably, considering only full-length proteoforms or the protein accessions instead of the proteoforms, a similar picture of the best combinations

regarding the number of identifications was observed (Supplementary Fig. 16).

## Physicochemical properties and modifications

The various sample preparations led to the identification of proteoforms in different mass ranges (Fig. 5a). The highest number of proteoforms smaller than 10 kDa were identified in the ACN-depleted samples; however, in agreement with the literature, only a few proteoforms larger than 20 kDa were identified[41]. In contrast, the highest number of large proteoforms were identified using MWCO filters, with 282 proteoforms above 20 kDa. Notably, many of the investigated strategies for isolating suitable proteoforms resulted in identifying a similar number of large proteoforms compared to the 1D-LC approach, which analyzed the entire proteome.

After 1D-LC, GELFrEE, MWCO, SPE and SEC, the identified proteoforms showed a similar distribution of their pI (Fig. 5b). In contrast, many more proteoforms with an acidic pI were identified after sample preparation with PEPPI. Conversely, a bias toward proteoforms with an alkalic pI was observed after ACN depletion. One explanation for these observations could be the different pH values of the used solutions: the ACN depletion is performed in an acidic solution; thus, proteoforms with an acidic pI have a low net charge and are less soluble, potentially leading to a loss of these proteoforms. In contrast, in the PEPPI protocol, the proteoforms are extracted from the gel using an alkaline solution, resulting in a reversed effect. Furthermore, the MWCO filters and SEC resulted in a median GRAVY score of the proteoforms that was

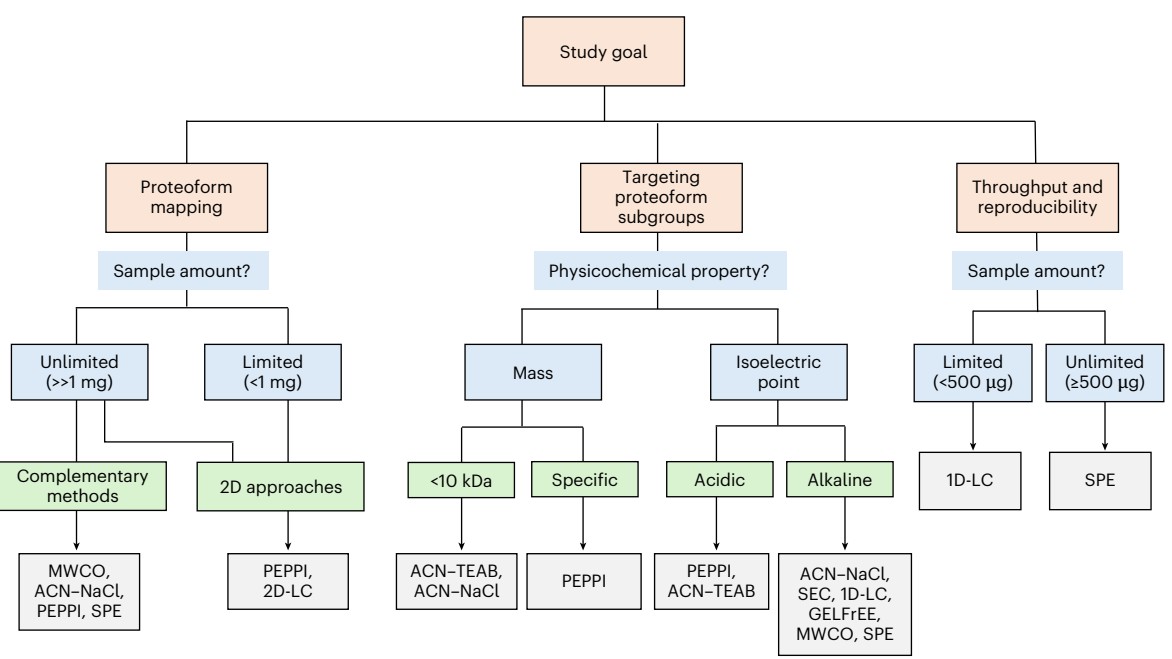

**Fig. 6 | Decision tree for the sample preparation of a qualitative TDP experiment in dependence on the research objective.** Specific examples to explain the use of the decision tree are provided in the Supplementary Information. Note that, for example, the selection of the number and combination of parallel approaches depends on the sample amount and instrument time available.

more negative than the other approaches, that is, more hydrophilic proteoforms were identified (Fig. 5c).

The various proteoform isolation approaches resulted in the identification of numerous modified proteoforms (Supplementary Table 3). Generally, the more identified proteoforms, the more modifications were assigned. We analyzed the datasets with MSTopDiff to detect the occurrence of artificially introduced modifications during sample preparation (Fig. 5d). Particularly in the MWCO dataset, a decent amount of oxidized mass features was detected, which may be attributed to the relatively long sample preparation time. In the PEPPI dataset, β-mercaptoethanol adducts were detected, originating from the sample incubation in Laemmli buffer before SDS–PAGE separation. The SPE sample preparation led to the detection of a mass shift assigned to formylation, which is due to the use of formic acid at room temperature.

Furthermore, we examined several variations of the investigated prefractionation approaches presented in the Supplementary Results and Supplementary Fig. 17.

### Multidimensional separation schemes

For comprehensive TDP studies, multidimensional separation schemes are typically used to increase the number of identifications[13]. To this end, the effects of an LC- and a gel-based separation approach on the identification of proteoforms were investigated using a recently developed two-dimensional low/low pH reversed-phase LC separation scheme[29] and the GELFrEE system[24], respectively (Supplementary Results).

Both approaches were used to separate proteoforms from human Caco-2 cells in eight fractions, resulting in the identification of a similar number of proteoforms (~2,150) and proteins (~770) (Supplementary Fig. 18a,b). Compared to the 1D-LC approach, the two-dimensional separation schemes resulted in more proteoform (~+145%) and protein (~+62%) identifications but required a substantially longer instrument time due to the analysis of eight fractions compared to one sample.

Notably, the replicates varied considerably regarding the number of identifications. However, the overlap coefficient between the replicates was high (~83% on protein and ~66% on proteoform level), demonstrating an overall good reproducibility. Approximately 18% of the identifications from both approaches were annotated proteoforms. Many truncated proteoforms originated from the hydrolysis of peptide bonds C terminal to aspartate residues (Supplementary Fig. 19). This observation agrees with the above-described isolation of small proteoforms using the GELFrEE system. Furthermore, the low/low pH LC-based fractionation used acidic conditions and has been shown to be susceptible to artificial hydrolysis of aspartic acid[29].

The LC-based fractionation identified a slightly higher number of large proteoforms (Supplementary Fig. 18c), which may be attributed to the full-proteome fractionation compared to the size-dependent fractionation by the gel-based approach. Furthermore, more alkaline and hydrophobic proteoforms were identified using the LC-based fractionation method compared to the gel-based approach (Supplementary Fig. 18d,e). Since the LC-based method was performed in low pH eluents, this observation is consistent with the bias against proteoforms with a pI value similar to the pH of the solution. In contrast, the gel-based separation was performed in more alkaline solutions. The fractionation efficiency of the two approaches is presented in the Supplementary Results and Supplementary Figs. 20 and 21.

### Proteoforms identified in this study

During this study, 257 LC–MS/MS runs were acquired, corresponding to a net instrument measuring time of approximately 27 days. Overall, 13,975 proteoforms from 2,720 proteins using a 1% context-dependent global false-discovery rate[48] were identified (Supplementary Tables 4–6), providing a comprehensive TDP dataset of human Caco-2 cells.

Approximately 14% (1,924) were full-length, and 86% (12,051) were truncated proteoforms, which is consistent with other TDP studies[4–6]. The average proteoform size was 10.3 kDa, with more than 470 proteoforms larger than 30 kDa (Supplementary Fig. 22). On average, the proteoforms were identified with a residue cleavage of 25%, with the smaller proteoforms having been identified with a higher residue cleavage. Several posttranslationally modified proteoforms were

identified, such as acetylation, phosphorylation, oxidation, methylation, butyrulation, trimethylation, myristylation and geranylation. Furthermore, many proteins were identified with multiple proteoforms carrying different or different localized PTMs (Supplementary Figs. 23–25). The most abundant proteoforms and a detailed analysis of the termini of the identified proteoforms are presented in the Supplementary Results and Supplementary Fig. 26.

## Discussion

We systematically investigated the influence of different sample preparation procedures on proteoform and protein identifications in TDP. The established new LC–MS/MS workflow, including optimized FAIMS–MS settings, number of replicates and injection amounts, ensured a fair comparison of the different sample preparation steps and can be applied to any TDP study.

We showed that each step in the sample preparation influences the number and the physicochemical properties of identifications, that is, biases toward specific proteoform subgroups, such as small, hydrophobic or acidic proteoforms, were introduced depending on the applied approaches (Supplementary Table 7). Moreover, we examined the proteoform quality regarding potential artificially introduced modification (for example, due to covalent and noncovalent adducts or peptide bond hydrolysis). While many sample preparation steps are similar in BUP and TDP, the impact of these steps is often overseen in BUP due to the need for protein inference. Thus, we suggest that the results of the intact protein level-centric TDP approach presented here can be used for optimizing BUP workflows, too.

In many TDP studies, including this one, MCW precipitation was used for sample cleanup to remove LC–MS incompatible compounds. Several studies have shown that sample cleanup considerably influences proteoform identification[47,49,50]. The application of SPE or MWCO allows direct sample cleanup, potentially omitting the precipitation, which may be particularly useful in cases where only limited starting amounts of biological material are available.

The findings described here allow us to recommend several guidelines for designing a qualitative TDP study, with a focus on sample preparation. Three main questions set the frame for the experimental design: (1) what are the objectives of the study? Common objectives are, for example, proteoform mapping, targeting a specific subgroup of proteoforms or reproducible identification of proteoforms with high throughput. (2) How much biological material is available, and what is the time needed for the analysis? (3) How to avoid sample loss and artifacts?

To select a sample preparation strategy meeting the demands of a given research question, Fig. 6 displays a decision tree, with some examples given in the Supplementary Notes. The isolation of proteoforms smaller than 30–50 kDa suitable for in-depth TDP can substantially increase the number of proteoforms and proteins compared to full lysate analyses. Depending on the available amount of biological material and instrument times, applying multiple orthogonal sample preparation techniques can considerably improve the depth of the analysis (Fig. 4e,f). Various sample preparation strategies introduce different biases for specific subgroups of proteoforms (Supplementary Table 7), enabling to set up a tailored combination of orthogonal strategies to improve proteoform coverage. However, our data show that the gain in identifications plateaus; thus, the number of combined approaches can be adapted to balance out the aimed depth of analysis and the time efforts and material needed. Further, our and many other studies[29,31] show that multidimensional fractionation schemes increase proteoform identifications, but at the cost of measurement time.

Independent of the analytical workflow chosen, some general recommendations applicable at all stages of the TDP sample preparation can be derived from our data, targeting the problem of sample loss and the prevention of artifacts (details and suggestions are provided in the Supplementary Notes and Supplementary Table 8): (1) heating and prolonged incubation times, especially in acidic solution, should be avoided as this may lead to hydrolysis of peptide bonds C terminal of aspartate residues; (2) the pH value of the cell lysis solution should be neutral or guided by the specific research question, with solution with acidic pH can effectively extract alkaline proteoforms and vice versa; (3) reduction and alkylation of proteoforms should be performed if there is no particular interest in assigning disulfides or other reversible cysteine modifications and (4) possible artificial modifications may be introduced by the cell lysis technique applied, the components of the protease inhibitor or other chemicals used in sample preparation; if possible, reagents that can lead to artificial modifications of proteoforms should be replaced.

In summary, this study provides a comprehensive overview of the influence of different commonly used sample preparation steps in TDP on proteoform and protein identifications. Each variation in the various sample preparation steps has distinct advantages and limitations, and the specific research objectives should guide the selection of sample preparation steps (Fig. 6 and Supplementary Tables 7 and 8). The data presented can help users make an informed decision on sample preparation based on the specific research objectives.

## Online content

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

## Methods

### Material

The cOmplete protease inhibitor was from Roche and Acrylamide/Bis solution was from BioRad. Human Caucasian colon adenocarcinoma (Caco-2) cells were from Sigma-Aldrich (cat. no. 86010202). The PBS, RPMI-1640 medium, fetal bovine serum and TrypLE Express Enzyme were from Thermo Fisher Scientific. If not otherwise stated, all other chemicals were from Sigma-Aldrich. Deionized water (18.2 MΩ cm$^{-1}$) was prepared by an arium611 VF system (Sartorius).

### Cell cultivation, lysis and protein determination

Human Caco-2 cells were maintained according to the recommendations of the European Collection of Cell Cultures. The cells were cultured at 37 °C and 5% CO$_2$ in RPMI-1640 medium (25 mM HEPES, 2 mM L-glutamine, 0.013 mM phenol red) supplemented with 10% (v/v) fetal bovine serum. When a confluence between 90 and 100% was reached, the cells were passaged by detaching with TrypLE Express Enzyme. Before collection, the cells were washed three times with PBS (200$g$, 5 min, 25 °C). The cells were stored at −70 °C until cell lysis and further use.

If not stated otherwise, Caco-2 cells were lysed in 1× PBS (pH 7.4) supplemented with 1× cOmplete protease inhibitor by ultrasonication (5 × 20 s on ice). After centrifugation (20 min, 21,000$g$, 4 °C), the protein concentration was determined by Pierce BCA assays (Thermo Fisher, Scientific) following the manufacturer's instructions.

### Investigation of the influence of the cell lysis

Human Caco-2 cells were lysed via sonification in (1) PBS (pH 7.4), (2) 8 M urea with 100 mM ABC (Urea–ABC, pH 8), (3) 8 M guanidinium hydrochloride (GndHCl, unbuffered), (4) 1% SDS with 15 mM Tris (SDS–Tris, pH 8.8) or (5) 76% ACN with 50 mM sodium chloride, 0.1% TFA (ACN–NaCl, pH around 2) or (6) 100 mM triethylammonium bicarbonate (ACN–TEAB, pH 8.5). A protease inhibitor was added to each lysis solution to minimize proteolytic activity after cell lysis. PBS, Urea–ABC, SDS–Tris and GndHCl are standard lysis solutions for proteome extraction, whereas the lysis with ACN has been initially developed for enriching small proteoforms[41]. After cell lysis, protein determination was performed by BCA assay (Thermo Fisher, Scientific), and proteoforms were purified by MCW precipitation, except the ACN-based lysis solutions, which were dried via lyophilization. All samples were prepared in triplicates, resuspended in LC–MS loading buffer, and injected twice using the LMW and HMW methods.

### MCW precipitation

For protein purification, MCW precipitation was used[9]. Cell lysates containing 100 or 500 µg of protein were made up to 125 µl with MilliQ and 400 µl of methanol, 100 µl of chloroform and 275 µl of MilliQ were added. The samples were vortexed and centrifuged for 10 min at 14,000$g$, 20 °C. The upper phase was removed and 600 µl of methanol was added. After centrifugation (10 min, 14,000$g$, 20 °C), the supernatant was removed, and the protein pellet was washed twice with 600 µl of methanol. The protein pellet was dried in a fume hood and stored at −20 °C until further processing.

### Proteoform reduction and alkylation

To investigate the influence of cysteine reduction or alkylation on proteoform identifications, Caco-2 cells were lysed and the proteoforms were aliquoted and subjected in triplicates to reduction with or without subsequent alkylation. In addition, an untreated sample was used as a control.

In detail, Caco-2 cells were lysed in 8 M GndHCl and 200 mM triethylammonium bicarbonate (pH 8.5) supplemented with 1× cOmplete protease inhibitor, as described above. The proteoforms (100 µg in 50 µl of lysis buffer) were reduced by adding 1.6 µl of 200 mM tris(2-carboxyethyl)phosphine and incubation at 50 °C for 1 h.

After cooling to room temperature, alkylation was performed by adding 3.3 µl of 375 mM iodoacetamide and incubating for 30 min at room temperature in the dark. After that, the proteoforms were purified by MCW precipitation.

### 1D-LC analysis

For the 1D-LC analysis, 100 µg of Caco-2 proteins were purified by MCW precipitation. The proteoforms were resuspended in 200 µl of LC–MS loading buffer (3% ACN, 0.1% TFA) by multiple pipetting steps, roughly vortexing and ultrasonication. Note that a considerable proportion of the pellet could not be dissolved and was visible as an insoluble precipitate. Before LC–MS/MS analysis, the samples were centrifuged for 20 min at 4 °C, 21,100$g$, to remove insoluble precipitates.

### MWCO filter

For the enrichment of proteoforms below approximately 30 or 50 kDa, appropriate MWCO filters were used (Amicon Ultra-0.5, regenerated cellulose, Ultracell). In brief, 500 µg of Caco-2 proteins were precipitated by MCW precipitation and resuspended in 500 µl of 8 M GndHCl. The MWCO filters were equilibrated by adding 500 µl of 8 M GndHCl and centrifugation for 15 min at room temperature, 14,000$g$. The filters were then transferred to a new reaction vial, and the sample was added. The filters were centrifuged for approximately 15 min at 14,000$g$ until 50 µl of the samples remained on the filter. The filter was centrifuged again after adding 8 M GndHCl to a total volume of 500 µl. The flowthrough, containing the enriched small proteoforms, was purified with an equilibrated 3 kDa MWCO filter. First, the sample was centrifuged at 14,000$g$ for 30 min, and the flowthrough was discarded (residue in the filter was approximately 50 µl). Next, the sample was washed twice with 100 mM TEAB (pH 8.5) on the filter. The residual containing the re-buffered proteoforms was transferred to a new 1.5-ml reaction tube, where the filter was inverted and centrifuged at 2,000$g$ for 3 min. The sample was lyophilized and solubilized in 50 µl of LC–MS loading buffer.

### PEPPI–MS

PEPPI–MS was performed as described previously, with slight modifications[19]. In brief, 500 µl Caco-2 proteins were purified by MCW precipitation and resuspended in 100 µl of Laemmli buffer (35% MilliQ, 25% glycerol, 2% SDS, 0.001% bromphenol blue, 5%, 62.5 mM Tris-HCl, pH 6.8). After incubation for 10 min at 50 °C, each 8 µl of the sample (40 µg) was separated on a self-casted SDS–PAGE gel (4% stacking, 16% separation gel) using a Tris-Glycine running buffer (25 mM Tris base, 192 mM glycine, 1% SDS). A prestained marker was used to monitor the separation of the proteoforms, and electrophoresis was stopped once separation of proteoforms <30 kDa was achieved (Supplementary Fig. 11a). Immediately after electrophoresis, one band below approximately 30 kDa of two adjacent lanes was excised based on the prestained protein marker without previous fixation or staining of the gel. The gel bands were crushed with a pestle, and the extraction solution (0.1% SDS, 100 mM ammonium bicarbonate buffer (ABC), pH ~8) was added. After incubation for 10 min at 20 °C and 1,500 rpm, the gel pieces were removed using a CoStar-X filter (2,000$g$, 3 min).

In the PEPPI-MCW protocol, the solution was subjected to MCW precipitation to purify the proteoforms. For the PEPPI-AnExSP approach, the samples were purified using an anion-exchange disk-assisted sequential sample preparation as previously described[49,51]. In brief, the solutions were subjected to 3-kDa MWCO filters and washed (13,500$g$, 25 min) twice with 8 M urea and 100 mM ABC. Subsequently, the proteoforms were purified by anion-exchange material via the stage-tip protocol. The stage tip was conditioned with 40 µl of methanol and equilibrated with 120 µl of 100 mM ABC (7,000$g$, 3 min). The sample was loaded on the stage tip and washed with 120 µl of 100 mM ABC. The proteoforms were eluted with 80 µl of 50% ethanol and 0.5% formic acid. The sample was lyophilized and solubilized in 20 µl of LC–MS loading buffer.

## GELFrEE

The proteoform fractionation by GELFrEE (Expedeon) was performed according to the manufacturer's recommendations. In brief, 500 μg of Caco-2 proteins were purified by MCW precipitation and resuspended in 30 μl GELFrEE loading buffer, 112 μl of MilliQ and 8 μl of 1 M dithiothreitol. The sample was incubated for 10 min at 50 °C before loading in the sample loading chamber. For the enrichment of proteoforms smaller than approximately 30 kDa, an 8% Tris-Acetate cartridge was used and the separation was performed according to the manufacturer's protocol. Only the first fraction containing proteoforms smaller than ~30 kDa was purified by MCW precipitation and resuspended in 50 μl of LC–MS loading buffer before LC–MS/MS analysis.

A 10% Tris-Acetate cartridge was used for proteome fractionation, and eight fractions below approximately 50 kDa were collected according to the manufacturer's recommendations. The success of the separation was validated by SDS–PAGE analysis and Coomassie-staining (Supplementary Fig. 11b). The fractions were purified by MCW precipitation and resuspended in 20 μl of loading buffer before LC–MS/MS analysis. Note that the production of the GELFrEE system and its cartridges has been discontinued and are, thus, no longer commercially available. Alternatively, self-casted GELFrEE cartridges can be used[52].

## SEC

SEC was performed on an high-performance liquid chromatography (HPLC) Ultimate 3000 (Thermo) system equipped with a Biosep-S3000 column (300 × 4.6 mm, Phenomenex). Approximately 500 μg of Caco-2 proteins were purified by MCW precipitation and resuspended in 50 μl of 8 M GndHCl before adding 150 μl MilliQ water. The sample was centrifuged and transferred in an LC vial. An isocratic flow of 40% ACN and 0.1% TFA at 300 μl min⁻¹ was used (30 °C). Before the separation, a protein standard mixture was separated on the SEC column to ensure that the LC and the column were in a good state (Supplementary Fig. 11c). For the enrichment of small proteoforms, 50 μl of the sample (125 μg) was injected, and one fraction between 8 and 11 min was collected (Supplementary Fig. 11d). The sample was lyophilized and resuspended in 20 μl of LC–MS loading buffer before LC–MS/MS analysis.

## SPE

SPE was performed as described previously, with minor modifications[26]. Here, 500 μg of Caco-2 proteins were purified by MCW precipitation and resuspended in 150 μl 8 M GndHCl. After adding 1 ml 5% formic acid, the sample was centrifuged for 20 min at 21,100$g$, and the supernatant was transferred into a new reaction vial. A C18 SepPak (1cc, 50 mg; Waters) or C4 Supra-Clean (1cc, 50 mg; PerkinElmer) cartridge was used for SPE. The SPE material was activated by 2× 1 ml 100% ACN and equilibrated by 2× 1 ml 5% formic acid before sample loading. The sample was washed twice with 5% formic acid and eluted with 300 μl of 70% ACN and 300 μl of 100% ACN. Before lyophilization, 600 μl of MilliQ was added, and the sample was frozen at −80 °C. The proteins were resuspended in 50 μl of LC–MS loading buffer before LC–MS/MS analysis.

## ACN depletion

Enriching small proteoforms by ACN depletion was performed as described previously, with slight modifications[41]. In brief, 500 μg of Caco-2 proteins were purified by MCW precipitation, and 76% ACN supplemented with 50 mM sodium chloride, 0.1% TFA (rough pH 2, acidic conditions) or 100 mM TEAB (pH 8.5, alkaline conditions) were added. The small protein fraction was resuspended by vigorous vortexing, ultrasonication and incubation for 1 h at 20 °C and 1,500 rpm. The samples were centrifuged (20 min, 21,100$g$, 20 °C) to remove the insoluble (large) proteoforms, and the supernatant was transferred in a new reaction vial. Before lyophilization, 100 μl of MilliQ was added and the sample was frozen at −80 °C. The proteins were resuspended in 20 μl of LC–MS loading buffer before being subjected to LC–MS/MS analysis.

## Reversed-phase low pH fractionation

Fractionation of proteoforms by reversed-phase chromatography was performed on a Dionex Ultimate 3000 HPLC system (Thermo Fisher Scientific), equipped with a PLRP-S column (8 μm, 1,000 Å, 2.1 × 150 mm (Agilent)), as previously described[29]. In brief, 500 μg Caco-2 proteins were purified by MCW precipitation and resuspended in 40 μl of 8 M GndHCl before adding 50 μl of eluent A (0.1% TFA). Then 70 μl of the sample (~350 μg) was injected, and the proteoforms were separated (flow rate 300 μl min⁻¹) over a 60 min gradient from 25 to 65% eluent B (80% ACN, 0.1% TFA). The separation was monitored by an ultraviolet-visible light detector ($λ = 214$ nm) connected to the column outlet. A protein mixture was separated before complex proteoform separation to ensure that the column and LC were in a good state (Supplementary Fig. 11e). Fractions were fractionated manually in a reaction vial filled with 300 μl of 100 mM TEAB (pH 8.5): fraction 1, 3–8 min; fraction 2, 8–13 min; fraction 3, 13–15 min; fraction 4–47, each minute and fraction 48, 59–63 min (Supplementary Fig. 11f). The fractions were vacuum-dried by lyophilization and resuspended in 40 μl of LC–MS loading buffer. A concatenation strategy was applied, with a total of eight resulting pools. Pool A consisted of fractions 1, 9, 17, 25, 33 and 41; pool B of fraction 2, 10, 18, 26, 34, 42 and so on. The pools were dried by lyophilization and resuspended in 20 μl of LC–MS loading buffer before LC–MS/MS analysis.

## LC–MS/MS analysis

Before mass spectrometric analysis, the proteoforms were separated using a Dionex U3000 UHPLC system (Thermo Fisher Scientific) equipped with a C4 column (50 cm × 75 μm, 2.6 μm, 150 Å, Thermo Fisher Scientific). A C4 precolumn (C4 PepMap300, 5 μm, 300 Å, Thermo Fisher Scientific) was exploited for sample loading (30 μl min⁻¹ of 3% ACN, 0.1% TFA). The separation was performed with eluent A (0.05% formic acid) and eluent B (0.04% formic acid in 80% ACN) using a 120-min gradient, with a flow rate of 300 nl min⁻¹ and a temperature of 45 °C: 0–5 min 4% B, 5–7 min 4–15% B, 7–127 min 15–60% B, 127–129 min 60–90% B, 129–140 min 90% B, 140–140.1 min 90–4% B, 140.1–150 min 4% B. The LC was coupled online to a Fusion Lumos Tribrid mass spectrometer (Thermo Fisher Scientific, operated with the Thermo Scientific Orbitrap Tribrid Series 3.4 instrument control application, v.3.4.3072.18) equipped with the FAIMS Pro Interface. All samples were injected twice using two different MS methods with internal compensation voltage stepping and optimized MS settings to target the low/medium-molecular-weight range (LMW method) and the HMW range (HMW method)[32]. The LMW method used the compensation voltages −60, −50, −40 and −25 V and was performed in 'peptide mode'. Within a mass range of 400–1,800 $m/z$, MS1 spectra with a resolution of 60,000 or 120,000 (compensation voltages −60, −50 or −40, −25 V), an automatic gain control (AGC) target of 200%, a maximal injection time of 200 or 250 ms and two or four microscans were acquired. Fragment spectra were acquired with a 3-s cycle time per compensation voltage, and the most intense ions (dynamic exclusion enabled, $n = 2$ within 30 s, 60 s exclusion, ±1.5 $m/z$ tolerance) were fragmented using collision-induced dissociation (CID) with a normalized collision energy of 25%. The MS2 settings were: 150–2,000 $m/z$ mass range, 50,000 or 60,000 (compensation voltages −60, −50/−40, −25 V) resolution, 800 or 1,000% AGC target, 118 or 246 ms maximal injection time and two or four microscans. In contrast, the HMW method was acquired in 'protein mode' using the compensation voltages −30, −20, 0 and +15 V. Spectra were acquired using a medium/high acquisition strategy (resolution MS1 7,500, MS2 60,000)[53], and proteoforms were fragmented by electron-transfer higher-energy collisional dissociation (10 ms electron-transfer dissociation, 23% normalized collision energy HCD). The MS1/MS2 settings were the same for all compensation voltages: 200 or 1,000% AGC target, 50- or 250-ms maximal injection time and ten or six microscans. A tabular overview of the used MS settings is given in Supplementary Table 1.

## Proteoform identification

Proteoform identification was performed with ProSightPD (v.4.2) within the Proteome Discoverer Suite (v.3.0.0.757). Unless stated otherwise, default parameters for all processing and consensus nodes were used. The data acquired with high resolution on MS1 and MS2 levels (high/high data) were processed with the high/high cRAWler (Xtract deconvolution; signal to noise threshold 3, charge state 0–30, fit factor 0.44). The low-resolution MS1 and high-resolution MS2 were processed with the medium/high cRAWler (kDecon deconvolution; precursor mass 5,000–60,000, signal to noise result cutoff 1). The deconvolved spectra were searched against a reviewed human database downloaded from UniProt[54] as an XML file, including all known alternative splicing variants, signal- and pro-peptides, as well as cotranslational modifications and PTMs (taxon ID 9606, release 2023_01) using the Annotated Proteoform Search and Subsequence Search (note that artificially introduced adducts were not considered for database search to avoid a massive increase in the search space when searching for variable modifications). Protein N-terminal acetylation was included in the database search. The precursor tolerance was set to 10 ppm, except for the Annotated Proteoform Search of the medium/high data, where it was set to 2.2 Da. The fragment mass tolerance was set to 10 ppm for all searches. For CID fragmentation, ProSightPD considered b- and y-ions, and for electron-transfer higher-energy collisional dissociation fragmentation, b-, y-, c- and z-ions. The consensus step filtered the identifications using a context-dependent 1% false-discovery rate (that is, on the level of proteoform spectrum matches, isoforms, proteins and proteoforms)[48]. The processing steps were combined in a multiconsensus search for merging multiple datasets.

To increase the quality of the results, only proteoforms with a characterization score (C-score) higher than 40 and proteins with at least one proteoform with a C-score >40 were reported[36]. The C-score is a metric that defines well-characterized proteoforms and stringent filtering allows the minimization of false-positive proteoform identifications.

## Data analysis and quality control

In the Supplementary Notes, various observations that notoriously occur during sample preparation and LC–MS/MS analysis are presented, which can provide valuable insights into potential issues to consider when designing TDP experiments.

For deconvolution with FLASHDeconv (v.2.0, default settings)[42], the raw data were converted into mzML files using msConvert. MSTopDIFF (v.1.1.0, intensity × count histogram with a bin size of 0.01 Da) analysis was performed to detect artificial modifications[43]. The data evaluation was performed based on exported result files from Proteome Discoverer using in-house Python (v.3.11.2) scripts using pyOpenMS (v.3.1.0)[55] and Pyteomics (v.4.6)[56]. The overlap coefficient was calculated by dividing the total number of shared items between two datasets by the smaller dataset's length. The proteoforms were matched regarding their ProForma annotation[57]. The Python module Pyteomics was used to compute the pI and the grand average of hydropathy (GRAVY) score of the identified proteoform sequences (without considering PTMs)[56]. The GRAVY score is a measure for the hydrophobicity of the proteoforms: that is, the more negative the GRAVY score, the more hydrophilic the proteoform.

The term annotated proteoforms refers to full-length proteoforms deposited in the database, including start methionine excision, and previously described truncated proteoforms, for example, due to signal peptide cleavage. In contrast, subsequence proteoforms are all truncated ones not defined in the database. The potential truncation sites were determined by matching the truncated proteoform sequence to the full-length sequence deposited in the database.

## Reporting summary

Further information on research design is available in the Nature Portfolio Reporting Summary linked to this article.

## Data availability

All raw data and database search results have been uploaded to the ProteomeXchange Consortium via the PRIDE partner repository with the dataset identifier PXD049969 (ref. 58). The protein database can be downloaded from UniProt (https://www.uniprot.org, taxon ID 9606). Source data are provided with this paper.

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

## Acknowledgements

We thank M. K. Steinbach for cultivating the human Caco-2 cells, T. Matzanke and J. Rost for method optimization, and L. Cassidy and J. Kim for helpful discussion. This work was supported by the DFG Cluster of Excellence 'Precision Medicine in Inflammation', RTF V (to A.T.). The funders had no role in study design, data collection and analysis, decision to publish or preparation of the manuscript.

## Author contributions

A.T., P.T.K., K.J. and O.K. conceptualized the study. P.T.K. developed the methodologies, performed the experiments and LC–MS/MS measurements. P.T.K. and K.J. performed the data analysis. P.T.K. and A.T. wrote the original draft of the manuscript. All authors contributed to the writing, review and editing of the final manuscript.

## Funding

## Competing interests

The authors declare no competing interests.

## Additional information

**Correspondence and requests for materials** should be addressed to Andreas Tholey.

# Reporting Summary

## Statistics

For all statistical analyses, confirm that the following items are present in the figure legend, table legend, main text, or Methods section.

| n/a | Confirmed | |
|---|---|---|
| ☐ | ☒ | The exact sample size (*n*) for each experimental group/condition, given as a discrete number and unit of measurement |
| ☐ | ☒ | A statement on whether measurements were taken from distinct samples or whether the same sample was measured repeatedly |
| ☒ | ☐ | The statistical test(s) used AND whether they are one- or two-sided<br>*Only common tests should be described solely by name; describe more complex techniques in the Methods section.* |
| ☒ | ☐ | A description of all covariates tested |
| ☒ | ☐ | A description of any assumptions or corrections, such as tests of normality and adjustment for multiple comparisons |
| ☒ | ☐ | A full description of the statistical parameters including central tendency (e.g. means) or other basic estimates (e.g. regression coefficient) AND variation (e.g. standard deviation) or associated estimates of uncertainty (e.g. confidence intervals) |
| ☒ | ☐ | For null hypothesis testing, the test statistic (e.g. *F*, *t*, *r*) with confidence intervals, effect sizes, degrees of freedom and *P* value noted<br>*Give P values as exact values whenever suitable.* |
| ☒ | ☐ | For Bayesian analysis, information on the choice of priors and Markov chain Monte Carlo settings |
| ☒ | ☐ | For hierarchical and complex designs, identification of the appropriate level for tests and full reporting of outcomes |
| ☒ | ☐ | Estimates of effect sizes (e.g. Cohen's *d*, Pearson's *r*), indicating how they were calculated |

*Our web collection on statistics for biologists contains articles on many of the points above.*

## Software and code

Policy information about availability of computer code

| Data collection | Thermo Scientific™ Orbitrap™ Tribrid™ Series 3.4 instrument control application (v3.4.3072.18) |
|---|---|
| Data analysis | (i) ProSightPD (v4.2) within the Proteome Discoverer Suite (v3.0.0.757) (commercial, Thermo Scientific); (ii)FLASHDeconv (v2.0, default settings),Jeong et al., Cell Syst. 2020, 10 (2), 213-218.e6; (iii) MSTopDIFF (v1.1.0, default settings); Kaulich et al., J Proteome Res 2022, 21: 20-29; (iv) Python V11.3.2, including Pyteomics V4.6, and PyOpenMS V.3.1.0 |

For manuscripts utilizing custom algorithms or software that are central to the research but not yet described in published literature, software must be made available to editors and reviewers. We strongly encourage code deposition in a community repository (e.g. GitHub). See the Nature Portfolio guidelines for submitting code & software for further information.

## Data

Policy information about availability of data

All manuscripts must include a data availability statement. This statement should provide the following information, where applicable:
- Accession codes, unique identifiers, or web links for publicly available datasets
- A description of any restrictions on data availability
- For clinical datasets or third party data, please ensure that the statement adheres to our policy

ProteomeExchange Consortium via the PRIDE partner repository with the dataset identifier PXD049969. Human protein database including all known modifications for database search can be downloaded as XML file from UniProt (https://www.uniprot.org, taxon ID 9606).

## Human research participants

Policy information about studies involving human research participants and Sex and Gender in Research.

| Reporting on sex and gender | n/a |
|---|---|
| Population characteristics | n/a |
| Recruitment | n/a |
| Ethics oversight | n/a |

Note that full information on the approval of the study protocol must also be provided in the manuscript.

# Field-specific reporting

Please select the one below that is the best fit for your research. If you are not sure, read the appropriate sections before making your selection.

☒ Life sciences ☐ Behavioural & social sciences ☐ Ecological, evolutionary & environmental sciences

For a reference copy of the document with all sections, see nature.com/documents/nr-reporting-summary-flat.pdf

# Life sciences study design

All studies must disclose on these points even when the disclosure is negative.

| Sample size | Cell culture based experiments: one batch (biological replicate) of Caco-2 cells was used for all experiments performed in this study. For the optimization of the LC-MS workflow, the influence of the number of replicate measurements on the identifications were investigated. Three replicates were a compromise to cover a high number of proteoforms and to investigate the reproducibility while being economical with measurement time.<br><br>For investigating the influence of sample preparation (i.e., cell lysis, enrichment of suitable proteoforms, proteoform fractionation), the respective sample preparation was independently performed three times. |
|---|---|
| Data exclusions | only protein/proteoform identifications matching the quality criteria/tresholds given in the manuscript are reported (1% false discovery rate; C-score >40) |
| Replication | Three technical replicates were performed as standard, unless otherwise stated. All technical replicates performed were included in this study, unless a LC-MS run was interrupted due to technical issues (i.e., instrument performance); in the latter case, the entire measurement series was discarded and repeated. |
| Randomization | No randomization was necessary as this was a qualitative study, based on the performance of technical replicates out of singe biological replicate (single batch cell culture). |
| Blinding | No blinding was necessary as this was a qualitative study, based on the performance of technical replicates out of singe biological replicate (single batch cell culture). |

# Reporting for specific materials, systems and methods

We require information from authors about some types of materials, experimental systems and methods used in many studies. Here, indicate whether each material, system or method listed is relevant to your study. If you are not sure if a list item applies to your research, read the appropriate section before selecting a response.

### Materials & experimental systems

| n/a | Involved in the study |
|---|---|
| ☒ | ☐ Antibodies |
| ☐ | ☒ Eukaryotic cell lines |
| ☒ | ☐ Palaeontology and archaeology |
| ☒ | ☐ Animals and other organisms |
| ☒ | ☐ Clinical data |
| ☒ | ☐ Dual use research of concern |

### Methods

| n/a | Involved in the study |
|---|---|
| ☒ | ☐ ChIP-seq |
| ☒ | ☐ Flow cytometry |
| ☒ | ☐ MRI-based neuroimaging |

# Eukaryotic cell lines

Policy information about cell lines and Sex and Gender in Research

| | |
|---|---|
| Cell line source(s) | Human Caco 2 (colon adenocarcinoma) cells / European Collection of Cell Cultures |
| Authentication | the cell line was not authenticicated |
| Mycoplasma contamination | not tested |
| Commonly misidentified lines (See ICLAC register) | The cell line used is not listed in the ICLAC register (release 2024, April, 26). |

