## [Peer Review File · Nature Methods]

Influence of Different Sample Preparation Approaches on Proteoform Identification by Top-Down Proteomics

Corresponding Author: Professor Andreas Tholey

Version 0:

Decision Letter:

28th May 2024

Dear Professor Tholey,

Your Analysis, "Influence of Different Sample Preparation Approaches on Proteoform Identification by Top-Down Proteomics", has now been seen by 4 reviewers. As you will see from their comments below, although the reviewers find your work of considerable potential interest, they have raised a number of concerns. We are interested in the possibility of publishing your paper in Nature Methods, but would like to consider your response to these concerns before we reach a final decision on publication.

We therefore invite you to revise your manuscript to address these concerns. In particular we ask you to focus on providing a clear message and practical advice for readers.

Link Redacted

We hope to receive your revised paper within 4 weeks. If you cannot send it within this time, please let us know. In this event, we will still be happy to reconsider your paper at a later date so long as nothing similar has been accepted for publication at Nature Methods or published elsewhere.

OPEN SCIENCE REQUIREMENTS

REPORTING SUMMARY AND EDITORIAL POLICY CHECKLISTS

IMAGE INTEGRITY

DATA AVAILABILITY

All novel DNA and RNA sequencing data, protein sequences, genetic polymorphisms, linked genotype and phenotype data, gene expression data, macromolecular structures, and proteomics data must be deposited in a publicly accessible database, and accession codes and associated hyperlinks must be provided in the "Data Availability" section.

CODE AVAILABILITY

Please include a "Code Availability" subsection in the Online Methods which details how your custom code is made available. Only in rare cases (where code is not central to the main conclusions of the paper) is the statement "available upon request" allowed (and reasons should be specified).

MATERIALS AVAILABILITY

ORCID

Nature Methods is committed to improving transparency in authorship. As part of our efforts in this direction, we are now requesting that all authors identified as 'corresponding author' on published papers create and link their Open Researcher and Contributor Identifier (ORCID) with their account on the Manuscript Tracking System (MTS), prior to acceptance. This applies to primary research papers only. ORCID helps the scientific community achieve unambiguous attribution of all scholarly contributions. You can create and link your ORCID from the home page of the MTS by clicking on 'Modify my Springer Nature account'. For more information please visit <http://www.springernature.com/orcid>.

Sincerely yours,
Allison

Allison Doerr, Ph.D.
Chief Editor
Nature Methods

Reviewers' Comments:

Reviewer #1:

Remarks to the Author:

Kaulich et al. present a very in-depth comparison of sample preparation protocols for top-down proteomics (TDP). Using human cell lysate (Caco-2 cells) as an example, they study in detail the effect of different sample preparation steps (lysis buffers/conditions, presence or absence of reduction and alkylation steps), and different enrichment and fractionation steps. The workflows are compared based on the number of protein and proteoform identifications, physicochemical properties, and modification profiles. In addition, single- and multidimensional fractionations are compared.

Overall, the study gives a very broad overview of different possible workflows in TDP. The combination of different conditions at several steps of the workflow would allow for an almost infinite number of comparisons, which is difficult to achieve in a manuscript such as this. Therefore, although the manuscript is very detailed, with many well-designed figures, the overall message remains somewhat unclear. Of course, different methods result in different identifications, and there is a certain overlap between different strategies. However, in my opinion, the authors would need to focus more on what, if any, more systematic conclusions can be drawn apart from the fact that the results are complementary. Some suggestions are given below.

In general, I consider the manuscript more suitable for a specialized audience, although it is a valuable resource for fine-tuning of TDP protocols. It does not resolve the main bottlenecks of TDP such as the very limited coverage of even medium-sized proteoforms or the low number of identifications per acquisition time.

Major comments:

Lines 114-116: The summary of the method performance appears very generic here: "highly reproducible", "the higher the amount, the higher the number and confidence ...". Please be more specific, also in other parts of the manuscript where similar statements are made.

Lines 154-156: Throughout the text, parameters that are changed for a certain evaluation step are interdependent on others, so it is sometimes difficult to draw conclusions. Here is an example: The ACN/TEAB buffer leads to identifications with lower pI. The authors conclude that this is because of poor solubility of proteins close to their pI. However, it could also be that the buffer extracts smaller proteins (with larger proteins being poorly soluble in ACN containing solutions), so that histones (as abundant proteins/proteoforms) are overrepresented in the sample, therefore shifting the mean pI. What I want to say here is that there may be more than one explanation for a certain observation, and the authors should consider that.

Line 269: In this section, multidimensional separation strategies are compared with each other, but I feel that the manuscript lacks an in-depth discussion of the improvement over 1D approaches, especially related to time investment.

Line 296: It would be good to have more information about FDR control for the entire project, when aggregating results over all datasets. The only relevant statement is given in the methods section (lines 573-574), where a 1% false discovery rate and a multi-consensus search are mentioned. At what level was the 1% FDR set? What does multi-consensus search mean exactly?

Line 490: For some workflows, different experimental conditions were (initially?) tested, e.g., for the SPE step here, where C4 and C18 materials were used. It is not clear which data was used for the results reported in Figures 4 and 5. Similar concerns may apply for other workflows - please ensure that it is clear which approach was used at all times.

Line 563: For data analysis, the authors state that default parameters were used unless otherwise noted. In some cases, it would still be more helpful to give more details. For example, which ion types were considered for searching EThcD data such as the one shown in SI Figure 4?

Minor comments:

Lines 123-124: For the "acidic acetonitrile-water solution", the presence of sodium chloride should not only be mentioned in the "abbreviation".

Lines 322: The authors mention non-covalent adducts here. Were they actually considered during data analysis? Supporting Information Tables 2 and 3 do not show any?

SI Figure 9: In panel E, I am curious what the mass shift to the left of single oxidation corresponds to as it seems to be fairly unique to this lysis buffer?

SI Figure 11: In the legend for panel C, first nine proteins, then six are mentioned.

SI Figure 21: References to panels D, E and F are missing. (Also, there is a typo, "one two eight" should read "one to eight".)

Reviewer #2:

Remarks to the Author:

In "Influence of Different Sample Preparation Approaches on Proteoform Identification by Top-Down Proteomics", Kaulich et al. provided a systematic comparison of several possible preparation methods for top-down proteomics. They examined the effects of different cell lysis methods, different fractionation and enrichment strategies, and different reduction/alkylation protocols on the total number of proteoforms identified. Comprehensive proteoform identification in complex samples is a major challenge and limited proteoform coverage is a persistent problem. This work is important as it provides the first parallel comparison between these methods, which can be a valuable reference for the top-down proteomics community. Experiments are well described, and the conclusions drawn from their experiments are well supported. This manuscript is well-suited for Nature Methods, and we recommend acceptance with minor revisions.

Please consider the following comments as you compose a revised manuscript:

1. According to Table 1, probe sonication was used for cell lysis. The use of probe sonication often induces proteoform truncations. This may have contributed to the high number of truncated proteoforms observed. It is worth noting in the manuscript that bath sonication, as a less intense sonication technique, can be used instead to lower this experimental artifact.
2. In the discussion section, it is mentioned that elevated temperature during reduction may have caused truncated proteoforms. Based on Table 1, the reductions were performed using TCEP at 50°C. However, TCEP works effectively at room temperature. Therefore, unnecessary modifications caused by heating could be diminished using a lower temperature.
3. Besides cell lysis, enrichment and separation, sample cleanup is an essential step in the sample preparation process. However, the authors did not include discussions on cleanup methods. This is an area where important aspects of sample preparation could be overlooked. Previous studies have been conducted to compare different cleanup methods (citation 46). This should be discussed.
4. In the protocol of SPE, the authors used formic acid to wash the samples, resulting in proteoform formylation. Alternatively, TFA can be used to avoid such artifacts. In addition, it should be noted that pepstatin A, a component in the complete protease inhibitor used for cell lysis, could be retained by SPE due to its hydrophobicity, and it can potentially introduce a contaminate peak.
5. Following MWC precipitation, the proteins are resuspended in LC-MS loading buffer prior to LC-MS/MS analysis. However, using formic acid is a more common practice, which should be addressed in the manuscript.
6. It is worth discussing the use of high pH fractionation as a fractionation technique. It is commonly used and provides a

fractionation that is orthogonal to low pH RPLC.

7. It should be noted in the manuscript that GelFrEE is no longer commercially available.

Version 1:

Decision Letter:

Our ref: NMETH-AS55537A

23rd Jul 2024

Dear Andreas,

Thank you for submitting your revised manuscript "Influence of Different Sample Preparation Approaches on Proteoform Identification by Top-Down Proteomics" (NMETH-AS55537A). It has now been seen by the original referees and their comments are below. The reviewers find that the paper has improved in revision, and therefore we'll be happy in principle to publish it in Nature Methods, pending minor revisions to satisfy the referees' final requests and to comply with our editorial and formatting guidelines.

TRANSPARENT PEER REVIEW

ORCID

Sincerely yours,
Allison

Allison Doerr, Ph.D.
Chief Editor
Nature Methods

Reviewer #1 (Remarks to the Author):

With this revised version, the authors have appropriately addressed my comments by providing extended discussion in the main text and the SI. I have no major comments, but I noticed some typos in the new sections of text in the SI document:

Dithiothreitol is misspelled twice, once on page 8 and once on page 15.

On page 53, "site reactions" should read "side reactions".

Reviewer #2 (Remarks to the Author):

nice job on the revisions

Reviewer #5 (Remarks to the Author):

The revisions are acceptable.

Version 2:

Decision Letter:

23rd Sep 2024

Dear Andreas,

I am pleased to inform you that your Analysis, "Influence of Different Sample Preparation Approaches on Proteoform Identification by Top-Down Proteomics", has now been accepted for publication in Nature Methods. The received and accepted dates will be 26 February 2024 and 23 September 2024. This note is intended to let you know what to expect from us over the next month or so, and to let you know where to address any further questions.

Over the next few weeks, your paper will be copyedited to ensure that it conforms to Nature Methods style. Once your paper is typeset, you will receive an email with a link to choose the appropriate publishing options for your paper and our Author Services team will be in touch regarding any additional information that may be required. It is extremely important that you let us know now whether you will be difficult to contact over the next month. If this is the case, we ask that you send us the contact information (email, phone and fax) of someone who will be able to check the proofs and deal with any last-minute problems.

Please note that *Nature Methods* is a Transformative Journal (TJ). Authors may publish their research with us through the traditional subscription access route or make their paper immediately open access through payment of an article-processing charge (APC). Authors will not be required to make a final decision about access to their article until it has been accepted. [Find out more about Transformative Journals](https://www.springernature.com/gp/open-research/transformative-journals)

If you are active on Twitter/X, please e-mail me your and your coauthors' handles so that we may tag you when the paper is published.

You can now use a single sign-on for all your accounts, view the status of all your manuscript submissions and reviews,

access usage statistics for your published articles and download a record of your refereeing activity for the Nature journals.

Best regards,
Allison

Allison Doerr, Ph.D.
Chief Editor
Nature Methods

** Visit the Springer Nature Editorial and Publishing website at http://editorial-jobs.springernature.com?utm_source=ejP_NMeth_email&utm_medium=ejP_NMeth_email&utm_campaign=ejp_Nmeth for more information about our career opportunities. If you have any questions please click [here](mailto:editorial.publishing.jobs@springernature.com).**

Open Access This Peer Review File is licensed under a Creative Commons Attribution 4.0 International License, which permits use, sharing, adaptation, distribution and reproduction in any medium or format, as long as you give appropriate credit to the original author(s) and the source, provide a link to the Creative Commons license, and indicate if changes were made. In cases where reviewers are anonymous, credit should be given to 'Anonymous Referee' and the source. The images or other third party material in this Peer Review File are included in the article's Creative Commons license, unless indicated otherwise in a credit line to the material. If material is not included in the article's Creative Commons license and your intended use is not permitted by statutory regulation or exceeds the permitted use, you will need to obtain permission directly from the copyright holder.

Editors request

In particular we ask you to focus on providing a clear message and practical advice for readers.

Our Response:

We fully agree that a clear message and practical advices were slightly hidden in our original submission, and therefore really acknowledge this issue, which was also raised by reviewer #1 (please find the same answers as below to his/her issue in the section “Response to reviewer-#1”).

The following measures were taken:

- 1) We added specific and general points to be taken into account when planning a TDP-experiment to the Discussion-section. As parts of this novel discussion was part of the original one, this required a partial rewording of the discussion in order to avoid unnecessary repetitions.

All aspects described now in re-worded discussion are supported by the data a presented beforehand, so these changes did not alter the original scientific content nor were new data necessary

In order to keep the manuscript in the tight range of word-count allowed, the main text contains general terms /descriptions of important point to be taken into account for experimental planning, separated in two sections (albeit without subtitles) general advices to setup a sample preparation strategy and second, advices to prevent sample loss and artifacts, which are important considerations independent of the choice of the sample preparation strategy.

Parts of the original discussion were also shifted to the Supplementary information, new chapter (see next point, 2))

- 2) A more detailed description of the general advices stated in the main manuscript are now outlined in the new chapter “Guidelines for the Sample Preparation in TDP” placed in the Supplementary materials. This section contains both a more detailed description of potential sources of errors but also provides hints for alternatives (e.g., the use of other reagents), together with references where applicable.
- 3) Furthermore, we added another table to the Supplementary Information (Supplementary Table 8). In this table we address potential factors and measures to be taken to avoid artifacts or hints to select a suitable approach and mention potential alternatives. Readers can use this table also as a source for troubleshooting in case artefact are observed in their own experiments.
- 4) In addition, we added a new figure in the main text (new Figure 6), showing a decision tree for the selection of a suitable sample preparation strategy in dependence on the objective of a given TDP experiment. In order to keep the limit of six elements (tables/figures) allowed for “Analysis articles”, we moved the original Table 1 from the main text to the Supplementary Information (new Supplementary Table 7).

Reviewer #1:

Remarks to the Author:

Kaulich et al. present a very in-depth comparison of sample preparation protocols for top-down proteomics (TDP). Using human cell lysate (Caco-2 cells) as an example, they study in detail the effect of different sample preparation steps (lysis buffers/conditions, presence or absence of reduction and alkylation steps), and different enrichment and fractionation steps. The workflows are compared based on the number of protein and proteoform identifications, physicochemical properties, and modification profiles. In addition, single- and multidimensional fractionations are compared.

*Overall, the study gives a very broad overview of different possible workflows in TDP. The combination of different conditions at several steps of the workflow would allow for an almost infinite number of comparisons, which is difficult to achieve in a manuscript such as this. Therefore, although the manuscript is very detailed, with many well-designed figures, **the** overall message remains somewhat unclear. Of course, different methods result in different identifications, and there is a certain overlap between different strategies. However, in my opinion, the authors would need to focus more on what, if any, more systematic conclusions can be drawn apart from the fact that the results are complementary. Some suggestions are given below.*

In general, I consider the manuscript more suitable for a specialized audience, although it is a valuable resource for fine-tuning of TDP protocols. It does not resolve the main bottlenecks of TDP such as the very limited coverage of even medium-sized proteoforms or the low number of identifications per acquisition time.

Our Response:

We thank the reviewer for this valuable input. We fully agree that a clear message and practical advices were slightly hidden in our original submission, and therefore really acknowledge this issue, which was also raised by the Editor.

To address the reviewer's concern, which was also raised by the Editor, the following measures were taken:

- 1) We added specific and general points to be taken into account when planning a TDP-experiment to the Discussion-section. As parts of this novel discussion was part of the original one, this required a partial rewording of the discussion in order to avoid unnecessary repetitions.

All aspects described now in re-worded discussion are supported by the data a presented beforehand, so these changes did not alter the original scientific content nor were new data necessary

In order to keep the manuscript in the tight range of word-count allowed, the main text contains general terms /descriptions of important point to be taken into account for experimental planning, separated in two sections (albeit without subtitles) general advices to setup a sample preparation strategy and second, advices to prevent sample loss and artifacts, which are important considerations independent of the choice of the sample preparation strategy.

Parts of the original discussion were also shifted to the Supplementary information, new chapter (see next point, 2))

- 2) A more detailed description of the general advices stated in the main manuscript are now outlined in the new chapter “Guidelines for the Sample Preparation in TDP” placed in the Supplementary materials. This section contains both a more detailed description of potential sources of errors but also provides hints for alternatives (e.g., the use of other reagents), together with references where applicable.
- 3) Furthermore, we added another table to the Supplementary Information (Supplementary Table 8). In this table we address potential factors and measures to be taken to avoid artifacts or hints to select a suitable approach and mention potential alternatives. Readers can use this table also as a source for troubleshooting in case artefact are observed in their own experiments.
- 4) In addition, we added a new figure in the main text (new Figure 6), showing a decision tree for the selection of a suitable sample preparation strategy in dependence on the objective of a given TDP experiment. In order to keep the limit of six elements (tables/figures) allowed for “Analysis articles”, we moved the original Table 1 from the main text to the Supplementary Information (new Supplementary Table 7).

Major comments:

Rev-#1-1: Lines 114-116: The summary of the method performance appears very generic here: "highly reproducible", "the higher the amount, the higher the number and confidence ...". Please be more specific, also in other parts of the manuscript where similar statements are made.

Our Response:

Due to the restricted word count in the main manuscript, we originally decided to make general statements rather than mentioning numbers in the main part; the exact numbers were then stated in the particular Supplementary Results/Figures. We agree that this led in some case to oversimplification or generic description. We therefore now added some exact numbers, or do explicitly refer to the particular Suppl. Figure throughout the manuscript.

Rev-#1-2: Lines 154-156: Throughout the text, parameters that are changed for a certain evaluation step are interdependent on others, so it is sometimes difficult to draw conclusions. Here is an example: The ACN/TEAB buffer leads to identifications with lower pI. The authors conclude that this is because of poor solubility of proteins close to their pI. However, it could also be that the buffer extracts smaller proteins (with larger proteins being poorly soluble in ACN containing solutions), so that histones (as abundant proteins/proteoforms) are overrepresented in the sample, therefore shifting the mean pI. What I want to say here is that there may be more than one explanation for a certain observation, and the authors should consider that.

Our Response:

We fully agree that some parameters are interdependent on each other, and due to the complexity of the proteome sample, the poor solubility of proteins close to their *pI* may be only one possible explanation in the given example.

We re-worded the sentence and added one additional sentence to clarify that the lower solubility of the proteoforms is only one possible explanation for the bias towards more basic proteoforms identified after ACN/NaCl lysis. Furthermore, we re-worded a sentence in the chapter "Physicochemical Properties and Modifications" to clarify that the suggested explanation is only one possible explanation, amongst others.

Rev-#1-3: Line 269: *In this section, multidimensional separation strategies are compared with each other, but I feel that the manuscript lacks an in-depth discussion of the improvement over 1D approaches, especially related to time investment.*

Our Response:

We thank the reviewer for this suggestion. We added a sentence to the chapter Results describing that the 2D approach led to a significant increase in proteoform identifications, however, on cost of increased instrument time. Of course, the latter is dependent on the number for fractions collected in first dimension, and we refer to the setup we used in our study (8 fraction = 8 times longer instrument time). This aspect is also discussed in new Discussion.

Rev-#1-4: Line 296: *It would be good to have more information about FDR control for the entire project, when aggregating results over all datasets. The only relevant statement is given in the methods section (lines 573-574), where a 1% false discovery rate and a multi-consensus search are mentioned. At what level was the 1% FDR set? What does multi-consensus search mean exactly?*

Our Response:

ProSightPD uses context-dependent FDR rates on the level of proteoform spectrum matches, proteins, isoforms, and proteoforms. ProSightPD divides the database workflow into the processing workflow to assign PrSMs and the consensus workflow for proteoform filtering (e.g., applying FDR cutoff). Multiple processing workflows can be combined in one consensus workflow (= multi-consensus search) to ensure proper global FDR calculation and to generate a single database search result file.

We added information about the applied 1% global FDR in the main manuscript (chapter "Proteoforms Identified in this Study") and to the online methods. Furthermore, we included the relevant reference (LeDuc et al., Mol. Cell. Proteomics 2019, 18 (4), 796–805).

Rev-#1-5: Line 490: *For some workflows, different experimental conditions were (initially?) tested, e.g., for the SPE step here, where C4 and C18 materials were used. It is not clear which data was used for the results reported in Figures 4 and 5. Similar concerns may apply for other workflows - please ensure that it is clear which approach was used at all times.*

Our Response:

We thank the reviewer for pointing this out. In the main manuscript, the described workflow for the SPE protocol was performed with C18 material, for the MWCO protocol with 30 kDa filter, for the PEPPI protocol with subsequence methanol-chloroform-water precipitation, and for the acetonitrile depletion with sodium chloride and TFA.

We added the specifications for the different experimental conditions in the chapter "Influence of Proteoform Enrichment, Depletion, and Purification". Furthermore, we included the specifications in the captions of the relevant figures in the manuscript and the Supplementary Information.

Rev-#1-6: Line 563: For data analysis, the authors state that default parameters were used unless otherwise noted. In some cases, it would still be more helpful to give more details. For example, which ion types were considered for searching EThcD data such as the one shown in SI Figure 4?

Our Response:

We included information about the ion types ProSightPD considers for CID and EThcD fragmentation and essential parameters for database searches in the Online Methods section. In addition, we added information to the captions of the relevant Suppl. Figures that clarify the considered fragment ions and explain the brackets in the fragment maps.

Minor comments:

Rev-#1-7: Lines 123-124: For the "acidic acetonitrile-water solution", the presence of sodium chloride should not only be mentioned in the "abbreviation".

Our Response:

We added "containing sodium chloride".

Rev-#1-8: Lines 322: The authors mention non-covalent adducts here. Were they actually considered during data analysis? Supporting Information Tables 2 and 3 do not show any?

Our Response:

The proteoform quality regarding artificially introduced modifications was determined by deconvolution with FLASHDeconv and analysis with MStopDiff, enabling the global detection of covalent and non-covalent (e.g., salt or SDS adducts) modifications. However, the detected artificially introduced modifications and non-covalent adducts were not included in the database search because they would have to be searched for as variable modifications, significantly increasing the search space (combinatorial explosion).

We added a sentence clarifying that the detected artificially introduced adducts cannot be considered for database search due to the significant increase in search space when searching for variable modifications to the Online methods section.

Rev-#1-9: *SI Figure 9: In panel E, I am curious what the mass shift to the left of single oxidation corresponds to as it seems to be fairly unique to this lysis buffer?*

Our Response:

Supplementary Figure 9E shows the MStoDiff analysis of the data acquired after the lysis with ACN/NaCl. Left to the peak assigned the single oxidation, a mass shift of 14.01 Da, and left to the double oxidation, a mass shift of 28.01 Da can be observed. In our experience, these mass shifts usually do not represent proteoform modifications but are artifacts caused by singly- or doubly-charged contaminations of unknown origin in the chromatogram. Manual inspection of randomly selected mass signals in the raw data did not reveal signs of covalently modified, e.g., methylated or dimethylated proteoforms. For example, for the latter, the MStoDiff derived value of 28.01 Da (bin size resolution is 0.01 Da) does not fit to this modification. We also observed this phenomenon in several earlier studies.

We included this information in the legend of Supplementary Figure 9E. Further, we added the occurrence of the 14.01/28.01 signal in MStoDiff and also in our “Supplementary Notes: Various observations”. We also added a note regarding the recognition of frequently observed PEG-contamination by using MStoDiff.

Rev-#1-10: *SI Figure 11: In the legend for panel C, first nine proteins, then six are mentioned.*

Our Response:

We deleted "six".

Rev-#1-11: *SI Figure 21: References to panels D, E and F are missing. (Also, there is a typo, "one two eight" should read "one to eight".)*

Our Response:

We added the references for D-F and corrected the typo.

Reviewer #2:

Remarks to the Author:

*In "Influence of Different Sample Preparation Approaches on Proteoform Identification by Top-Down Proteomics", Kaulich et.al provided a systematic comparison of several possible preparation methods for top-down proteomics. They examined the effects of different cell lysis methods, different fractionation and enrichment strategies, and different reduction/alkylation protocols on the total number of proteoforms identified. Comprehensive proteoform identification in complex samples is a major challenge and limited proteoform coverage is a persistent problem. This work is important as it provides the first parallel comparison between these methods, which can be a valuable reference for the top-down proteomics community. Experiments are well described, and the conclusions drawn from their experiments are well supported. This manuscript is well-suited for Nature Methods, and we recommend acceptance with **minor revisions**.*

Please consider the following comments as you compose a revised manuscript:

Rev-#2-1: 1. *According to Table 1, probe sonication was used for cell lysis. The use of probe sonication often induces proteoform truncations. This may have contributed to the high number of truncated proteoforms observed. It is worth noting in the manuscript that bath sonication, as a less intense sonication technique, can be used instead to lower this experimental artifact.*

Our Response:

We thank the reviewers for highlighting this point. We added a sentence highlighting the potential induction of artifacts during sonication, e.g., due to heating or mechanical stress, resulting in artificially truncated proteoforms or unwanted protein precipitation (reference added) to the new chapter "Guidelines for the Sample Preparation in TDP" in the Supplementary Information. Moreover, we highlight that less intense cell lysis techniques, such as using a sonication bath, should be considered.

Rev-#2-2: 2. *In the discussion section, it is mentioned that elevated temperature during reduction may have caused truncated proteoforms. Based on Table 1, the reductions were performed using TCEP at 50C. However, TCEP works effectively at room temperature. Therefore, unnecessary modifications caused by heating could be diminished using a lower temperature.*

Our Response:

We agree with the reviewers. Indeed, although several TDP studies performed TCEP reduction at elevated temperatures, it has been described that TCEP can effectively reduce disulfide bonds at room temperature.

We added a sentence in the relevant chapter of the main text to highlight the possibility of avoiding elevated temperatures during proteoform reduction to prevent the generation of artificially truncated proteoforms. Furthermore, we included this point in the new chapter "Guidelines for the Sample Preparation in TDP" in the Supplementary Information.

Rev-#2-3: 3. Besides cell lysis, enrichment and separation, sample cleanup is an essential step in the sample preparation process. However, the authors did not include discussions on cleanup methods. This is an area where important aspects of sample preparation could be overlooked. Previous studies have been conducted to compare different cleanup methods (citation 46). This should be discussed.

Our Response:

We fully agree with the reviewers that sample cleanup is an essential step in sample preparation and can result in a significant loss of proteoforms.

We added a paragraph in the main text, discussing the influence of sample cleanup on proteoform identifications. Furthermore, we added new references, which investigated sample clean-up methods in detail. The issue of sample cleanup was furthermore taken up in the new chapter “Guidelines for the Sample Preparation in TDP” in the Supplementary materials, as well as in the new Supplementary table 8.

Rev-#2-4: 4. In the protocol of SPE, the authors used formic acid to wash the samples, resulting in proteoform formylation. Alternatively, TFA can be used to avoid such artifacts. In addition, it should be noted that pepstatin A, a component in the complete protease inhibitor used for cell lysis, could be retained by SPE due to its hydrophobicity, and it can potentially introduce a contaminate peak.

Our Response:

We fully agree that other ion-pairing reagents, such as TFA, can also be used to avoid artificially introduced formylation. In our data, we have not observed a pepstatin A contamination peak, which could be, for example, attributed to the methanol-chloroform-water directly after cell lysis (which has also been described to remove low molecular weight peptides). However, we fully agree that, in general, the ingredients of the protease inhibitor may be enriched during sample preparation and then might be observed as contamination peaks.

We added a sentence in the new chapter “Guidelines for the Sample Preparation in TDP” in the Supplementary materials as well as in the new Supplementary Table 8, highlighting the possibility of using other ion-pairing reagents instead of formic acid to avoid artificially introduced formylation. Moreover, we stated that the protease inhibitor's ingredients could lead to artificially modified proteoforms and may be enriched during sample preparation and observed as contamination peaks.

Rev-#2-5: 5. Following MWC precipitation, the proteins are resuspended in LC-MS loading buffer prior to LC-MS/MS analysis. However, using formic acid is a more common practice, which should be addressed in the manuscript.

Our Response:

We thank the reviewer for highlighting this point. We added a sentence in the new chapter “Guidelines for the Sample Preparation in TDP” in the Supplementary materials as well as in the new Supplementary Table 8, highlighting the common practice of using concentrated formic acid and subsequent dilution for resuspending proteins, and we added an appropriate reference.

Rev-#2-6: 6. *It is worth discussing the use of high pH fractionation as a fractionation technique. It is commonly used and provides a fractionation that is orthogonal to low pH RPLC.*

Our Response:

We fully agree that high pH fractionation is a valuable technique for orthogonal fractionation of intact proteoforms prior to low pH separation. Previously, we compared a low/low pH with a high/low pH separation scheme for proteoform separation and demonstrated an advantage of the low/low-pH strategy in terms of the number of identified proteoforms (Kaulich et al., Proteomics, 2024; doi: 10.1002/pmic.202200542). However, the techniques resulted in highly complementary proteoform identifications.

We added a sentence in the Main Text and additionally, a paragraph in the Supplementary Information in the "Proteoform Fractionation" section, highlighting the orthogonal separation with a high/low pH separation scheme and adding appropriate literature. Furthermore, we specified the two 2D-reversed-phase LC-based separation schemes (high/low pH and low/low pH) in the introduction.

Rev-#2-7: 7. *It should be noted in the manuscript that GelFrEE is no longer commercially available.*

Our Response:

We mentioned in Table 1, now shifted to the Supplementary information (= Suppl. Table 7) that the GELFrEE system and its associated cartridges are no longer commercially available. Additionally, we highlighted this by adding a sentence in the online method section.

Response to Reviewer-#1:

Reviewer #1:

Remarks to the Author:

With this revised version, the authors have appropriately addressed my comments by providing extended discussion in the main text and the SI. I have no major comments, but I noticed some typos in the new sections of text in the SI document:

Dithiothreitol is misspelled twice, once on page 8 and once on page 15.

Our response: was corrected.

On page 53, "site reactions" should read "side reactions".

Our response: was corrected.